# Functional Classification and Characterization of the Fungal Glycoside Hydrolase 28 Protein Family

**DOI:** 10.3390/jof8030217

**Published:** 2022-02-22

**Authors:** Fernando Villarreal, Nicolás Stocchi, Arjen ten Have

**Affiliations:** Instituto de Investigaciones Biológicas CONICET-UNMdP, Mar del Plata B7602AYJ, Argentina; fernandovillarreal@mdp.edu.ar

**Keywords:** protein family, polygalacturonase, functional redundancy and diversification

## Abstract

Pectin is a major constituent of the plant cell wall, comprising compounds with important industrial applications such as homogalacturonan, rhamnogalacturonan and xylogalacturonan. A large array of enzymes is involved in the degradation of this amorphous substrate. The Glycoside Hydrolase 28 (GH28) family includes polygalacturonases (PG), rhamnogalacturonases (RG) and xylogalacturonases (XG) that share a structure of three to four pleated β-sheets that form a rod with the catalytic site amidst a long, narrow groove. Although these enzymes have been studied for many years, there has been no systematic analysis. We have collected a comprehensive set of GH28 encoding sequences to study their evolution in fungi, directed at obtaining a functional classification, as well as at the identification of substrate specificity as functional constraint. Computational tools such as Alphafold, Consurf and MEME were used to identify the subfamilies’ characteristics. A hierarchic classification defines the major classes of endoPG, endoRG and endoXG as well as three exoPG classes. Ascomycete endoPGs are further classified in two subclasses whereas we identify four exoRG subclasses. Diversification towards exomode is explained by loops that appear inserted in a number of turns. Substrate-driven diversification can be identified by various specificity determining positions that appear to surround the binding groove.

## 1. Introduction

### 1.1. The GH28 Gene Family Encodes a Number of Different Enzymatic Activities Directed at the Breakdown of the Major Plant Cell Wall Component Pectin

Pectin is one of the major compounds of the plant cell wall and therewith an important target for both pathogens and saprotrophs [1]. It is an acidic heteropolysaccharide that is deposited in the plant’s primary cell wall and middle lamella by which it provides the plant cell with mechanical strength, maintains its shape and provides the first barrier for many pathogens [1]. Pectins are important gelling components in the food industry and are used as filling material in pills [2]. Furthermore, as soluble fibre, they contribute to a healthy human diet [2].

In general, pectins form networks, either alone or with other cell wall components such as cellulose and lignin [1]. Pectin is divided into smooth regions, consisting of mostly homogalacturonan, and hairy regions, consisting of rhamnogalacturonan and xylogalacturonan. In addition, it can have various degrees of methylation and/or acetylation. Pectin with low levels of methylation and acetylation is also referred to as pectate or pectic acid [1]. Many different enzymes act on pectin; this is related to the fact that pectin forms an amorphous polysaccharide. Pectinasases are classified in the comprehensive BRENDA enzyme database [3] as well as in the more dedicated CAZy resource [4].

To understand the large variety of pectinolytic enzymes, we must have a look at the structure of pectin. Homogalacturonan consists of linear chains of α (1–4)-linked D-galacturonic acid that can be methylated at O-6 or acetylated at O-2 or O-3. It is subject to various enzymatic degradations. Demethylation of pectin is performed by pectin methylesterase (PME; EC3.1.1.11; CAZy CE8) and its deacetylation by pectin acetylesterase (PAE; EC3.1.1.6; CAZy CE12). It can be degraded directly by pectin lyase (PNL; EC4.2.2.10; CAZy PL1), whereas as pectate, it can be degraded by endo- and exoforms of pectate lyase (endoPL; EC4.2.2.2; CAZy PL1,2,3,9 or 10, and exoPL; EC4.2.2.9; CAZy PL1 or 9, respectively) as well as by endo- and exo-forms of polygalacturonases (endoPG; EC3.2.1.15; CAZy GH28, and exoPG; EC3.2.1.67; CAZy GH28). In bacteria, a third class of PGs (EC3.2.1.82; CAZy GH28) has been shown to release digalacturonates [5]. For more details see a recent review on pectinolytic enzymes [6].

Xylogalacturonan forms the most simple hairy pectin. It has xylose moieties attached to the O-3 through a β-1,3-glycosidic bond. Different types of xylose side chains can be attached, apparently depending on the species [6]. The side chain can be removed by β-xylosidase (EC 3.2.1.37; CAZy GH43) whereas endoxylogalacturonases (endoXG; EC3.2.1.-; CAZy GH28) can degrade the backbone, thereby releasing xylosylated oligogalacturonides [7]. It has also been shown that PGXA, PGXB and PGXC, which are considered exoPGs, have exoXG activity [8].

The more complex rhamnogalacturonan is further classified into RGI and RGII, the latter being very resistant to enzymatic breakdown. In RGI the galacturonan backbone is intercalated by L-rhamnose to which various types of arabinan, galactan and arabinogalactan can be attached at various positions [9]. Things are more complicated for rhamnogalacturonases (RGs), which have been named accordingly as endo- and exoRGs, whereas for the endoRG activity, two types of enzymes must be considered. Rhamnogalacturonan α-1,2-galacturonohydrolase (exoRG; EC3.2.1.173) releases D-galacturonic acid and is, as is endorhamnogalacturonase (endoRG, EC3.2.1.171) also part of the GH28 superfamily. Rhamnogalacturonan α-L-rhamnohydrolase (EC3.2.1.174) releases β-L-rhamnose and is part of the GH78 protein family. RGI is also subject to degradation by rhamnogalacturonan endo-lyase (EC4.2.2.23; CAZy PL4, 9, 11); rhamnogalacturonan exolyase (EC4.2.2.24; CAZy PL26) and rhamnogalacturonan acetylesterase (EC3.1.1.86, not classified by CAZy).

GH28 appears as an important and versatile protein superfamily that contains enzymes with substrate specificity for homogalacturonan (PGs), rhamnogalacturonan (RGs) and xyloglacturonan (endoXGs). For both PG and RG GH28 enzymes, endo- and exomodes have been described. Certain endoPGs show processive enzyme activity [10], which appears to be the standard mode for endoXG [11]. In addition, it has been reported that different PGs degrade pectins with different degrees of methylation [12]. This raises the question of how all these different functions have evolved in a single gene family.

The taxonomic distribution of GH28 enzymes is wide, but nevertheless restricted. Parsimony would dictate GH28 enzymes have originated in plants since that is where its substrate occurs. Interestingly, the GH28 protein superfamily is, in its turn, part of the even larger protein family of pectin lyase-like proteins (for information on this superfamily see the websites of Interpro [13], Superfamily [14] and SCOP [15]). These include many of the aforementioned enzymes, and also iota-carrageenase that degrades carrageenan [16], a major cell wall component of marine red algae. Indeed, phylogenetic analysis has identified there were at least five PG homologues in lycophyte *Selaginella moellendorffii* [17]. Plants have many PG paralogues but these have not yet been studied in much depth. Surprisingly, as compared to fungi, the plant GH28 repertoire seems restricted to PGs, although plants do have pectate lyases and also many pectin methyl esterases [17]. Hence, PGs and PLs likely evolved from a common ancestor that had the pectate lyase fold and was involved in the regulation and development of the ancestral plant cell wall. These, then, likely spread from plants to their pathogens and other symbionts as well as to saprotrophs, which live on decayed plant material. It is not known how all the different subfamilies in bacteria and particularly fungi have evolved. PGs have also been described in, albeit to a much lower extent, oomycetes [18], insects [19] and nematodes [20]. Humans lack PGs but might benefit from PGs present in their microbiome [21], although this has not been studied in much detail. Bacterial pectinases identified in sheep have been shown to act optimally at 40 °C, the ruminal temperature in sheep [22], suggesting the importance of pectinases in a plant-based diet. Altogether, it seems that GH28s are widespread but restricted by natural selection.

The fact that the GH28 family has not diversified as much in plants as it has in fungi is also explained by natural selection. The functions of plant PGs are sought in plant cell wall development, including loosening of this structure during fruit ripening and abscission of leaves. In pathogens, they function in infection. Although this has been shown unequivocally in only a few cases (e.g., [23,24,25,26]), there is sufficient evidence to sustain that they are involved in the infection process. Certainly, not all PGs are virulence factors, which can be partially explained by functional redundancy, given the often large numbers of paralogues. Much less is known regarding other GH28 subfamilies, and there is a certain consensus that many of these will somehow be involved in pathogenesis, but also that most cannot be considered as a virulence factor. For other organisms, such as saprotrophs, GH28 enzymes are involved in nutrient acquisition, which includes, among other processes, the uptake of galacturonic acid and subsequent degradation GH28 enzymes are involved in nutrient acquisition, which includes, among other processes, the uptake of galacturonic acid and subsequent degradation [27]. The D-galacturonic acid breakdown pathway is also fully functional in the necrotrophic pathogen *Botrytis cinerea* [28]. Surprisingly, it has also been shown that *B. cinerea* mutants in that pathway show wild type virulence, suggesting that PGs function in cell wall maceration rather than nutrient acquisition in this necrotroph.

GH28 and other pectinolytic enzymes are often referred to as cell wall degrading enzymes (CWDEs), focussing on their function for pathogens and other symbionts. An important aspect in plant-pathogen interactions is the effector-induced immune response. As such, CWDEs are considered as, or produce, so-called pathogen-associated molecular patterns (PAMPs). An endoPG purified from necrotrophic leotiomycete *Sclerotinia sclerotiorum*, but not its oligogalacturonide products, triggers hallmarks of apoptosis-type cell death. For closely related *B. cinerea*, it was shown that at least five endoPGs induce necrosis [25]. Furthermore, inactivation of BcPG2 by mutating one of the active aspartates resulted in the loss of necrotizing activation, suggesting activity and possible oligogalacturonic acids (OGAs), rather than the enzyme itself, to be involved as PAMP. Interestingly, plants have PG Inhibiting Proteins (PGIPs) and their action may be twofold. Primarily they inhibit PGs and therewith the degradation of the plant’s protective cell wall. Secondly, PGIPs may be involved in determining the length of OGAs and may therefore affect the plant defence response. The evasion of PGIP interaction as part of the arms race provides yet another functional constraint that acts on fungal GH28 enzymes.

### 1.2. Structural Analysis of GH28 Enzymes Shows Its Major Characteristics

Structures of GH28 homologues have been instrumental in the elucidation of the catalytic mechanism. Figure 1 shows a number of cartoons of structurally aligned, fungal GH28 enzymes as well as sequence conservation logos of the region that contains the catalytic residues (Figure 1E). The first GH28 structure that was resolved was that of an endoRG from *Aspergillus aculeatus* (PDB identifier 1RMG [29]) and the similarity it has to a pectate lyase made the authors follow the nomenclature that was used for that structure. It has four parallel β-sheets PB1, PB1a, PB2 and PB3 that together form a β-helix. Each full helix can be seen as a stack or slice, of which 1RMG has 14. PB1a is not present, or not described as such, in the pectate lyase structure. The helix forms a concave slide that appears to be able to hold a polysaccharide in its groove. The structure is supported by four disulphide bridges. SS1 sustains stack 1, SS2 stack 7, whereas SS3 and SS4 support stack 12 and 13, respectively. The first endoPG that was resolved is *Aspergillus niger* endoPGII (PDB identifier 1CZF [30]), which is more similar to pectate lyase such that it lacks PB1a. Furthermore, sheets PB2 and PB3 of stacks 9 to 12 combine to single continuous subsequences in β-sheet. Given the central role or importance of endoPGs, 1CZF is used as the major reference throughout this manuscript. Figure 1A shows a cartoon representation of 1CZF indicating the most important hallmarks.

Rhamnogalacturonase 1RMG was suggested to have a so-called inversion mechanism in which a carboxylate acts as a general acid that donates a proton to the glycosidic oxygen of the scissile bond [26]. A second carboxylate activates a water molecule that performs a nucleophilic attack on the anomeric carbon of the sugar. The authors suggested that Asp177, Asp180, Asp 197 and Glu198 are involved in this process. The publication of 1CZF was accompanied by mutational analyses [30]. Asp180 (Turn 3-6), Asp 201 and Asp 202 (corresponding to Asp177, Asp 197 and Glu198 of 1RMG, both on Turn 3-7) were shown to be required for catalysis [30]. Since endoPGs have a highly conserved His223 (Turn 3-8), this residue was also mutated which also resulted in an almost complete lack of activity. However, and although the Prosite PG pattern description [31] suggests this His to be catalytic for galacturonases, its absence in RGs suggest this residue is otherwise involved and does as such not take part in catalysis.

The turns that occur in between the different sheets are likely to be very important for catalysis. The active site is placed on the T3 turns of stacks 6, 7 and 8. Besides invoking a turn, T3-6 forms a flexible loop that is likely involved in the dynamics of substrate binding. In contrast, turn T3-7 is fixed to PB3-8 by means of disulphide bridge SS2. Gly224 was also suggested to be required since it has φ-ψ angles that are restricted to glycine. A relatively high content of glycine (13 within subsequence 171–231 (Figure 1E; 21% as compared to an estimated ~7% in protein [32]) is likely required for the dynamics of the complete reaction process.

A structural alignment of four fungal endoPG structures (PDB identifiers 1CZF, 1NHC, 1KCD [33] and 1HG8 [34]) shows, besides high overall conservation as indicated by the relatively low root-mean-square deviation (RMSD) values, that the catalytic residues and the residues that H bond to the substrate that was co-crystallized in 1KCD, appear highly conserved. Note that 1NHC, endoPGI from *A. niger*, is a so-called processive enzyme [10] that upon an initial endomode digestion proceeds in exomode. Additionally, the disulphide bridges are highly conserved, SS4 only being absent in 1KCD. Residues involved in the catalytic site, with subsite −1 and +1, are Ser121 (T3-4); Met150 (T3-5); His177 (T3-6); Ser229 (PB1-9); Arg256 and Lys258 (PB1-10); and Tyr291 on T1-11 (Figure 1C). The T1 and T3 loops, in general, can be envisaged to be involved in the H bonding of polygalacturonan, hence, will take part in the subsites. Although it has been suggested that endoPGs contain at least seven subsites [10,35] the actual subsites are not well defined. Pagès et al. [36] identified Asn186 at subsite −3, Met150 and Asp183 at subsite −2, Glu252, Asp282, and Gln288 at subsite +2 and finally Tyr326 at subsite +3 of endoPGII from *A. niger*, where −n corresponds to the non-reducing end and +n to the reducing end. Subsite variation will be larger than the amino acid variation of the actual catalytic site (subsites −1 and +1). Part of this variation is believed to be related to substrate specificity. The effect of methylation on PG activity has been studied and it is known that methylation at galacturonate residues next to the scissile bond severely hampers digestion, both in fungal [37] and bacterial [38] endoPGs. It is not known if these differences in activity and substrate specificity are conserved or if the versatility is simply provided by having many homologues.

Figure 1D shows structural alignments of the aforementioned endoRG (PDB identifier 1RMG) and a more recently resolved endoXG from *Aspergillus tubingensis* (PDB identifier 4C2L [9]) to endoPG 1CZF. Although particularly the endoRG shows a larger RMSD and has additional loops, the conservation of these enzymes shows a highly similar binding cleft and site, suggesting they share the catalytic mechanism. All four disulphide bridges are furthermore conserved. No structures for fungal exoPGs or exoRGs have been resolved.

The GH28 superfamily can be classified based upon actual activity, which would result in endo- and exoPGs, endo- and exoRGs as well as endoXG. Since many fungi have several paralogues for most of these classes, different additional suffixes have been used to identify different subfamily paralogues. This may be of importance if additional sequence diversification has indeed resulted in further functional diversification. EndoPGI 1NHC from *A. niger* is processive, whereas endoPGII 1CZF is not [10]. Both prefer pectate, whereas endoPGs C and E from *A. niger* prefer pectin with low degrees of methylation. EndoRGs and exoPGs in *Aspergilli* have been named as PGX using A, B, C. However, to the best of our knowledge, there has been no systematic analysis showing sub-subfamilies based on either sequence conservation or actual functional differences. Endo- and exoPGs have also been shown in bacteria and plants, but it is not known if these have a subfamily-specific common ancestor or whether these activities are the result of convergent evolution. There is knowledge regarding some specificity determining positions (SDPs) that define the three endo-mode subfamilies. Although we also know that substitution S91R determines the processive activity demonstrated by 1NHC with respect to 1CZF, there are other processive enzymes such as BcPG3 and BcPG6 from leotiomycete *B. cinerea* [25] that do not show the same SDP. Additionally, no SDPs have been demonstrated for exomode enzymes. Hence, although we have a lot of knowledge on a number of individual GH28 enzymes, we do not know how the GH28 family evolved specific traits such as how to deal with methylation and/or acetylation. Methylation and acetylation are often expressed in levels (e.g., percentage) but this results in an underestimation of the complexity of pectin diversity. The variety of substrates the various homologues can encounter hampers biochemical analyses. Then, we do not know why PGs and XG have His223 strictly conserved, whereas RGs lack this residue. We do not know how XG deals with its substrate. Although processivity is well understood for some PGs, it is not understood for other PGs. Related to processivity, a major challenge will be to explain exomode activity at the molecular level. No structures for fungal exoPGs have been resolved and there appear to be many different types of exoPGs with interesting substrate preferences [8]. Altogether, we lack both systematic knowledge on how the GH28 enzyme family has evolved in fungi, and how this has resulted in specific functionalities.

With the advent of almost every genome sequence, protein superfamilies can also be studied by computational biology. Computational biology can, among others, identify conserved sequence traits, which can give additional information on protein function. Protein sequence analysis and protein structure modeling [39] have developed a number of reliable methods for structure-function prediction. The construction of high-quality multiple sequence alignments is an ongoing area of research but has yielded a number of reliable aligners, among which MAFFT [40] provides a good trade-off between reliability and computational cost. FAMSA [41] is a more recent package that is very fast but, nevertheless, quite reliable. Phylogenetic reconstruction for protein superfamilies is performed by Maximum Likelihood (ML) methods such as PHYML [42] and FastTree [43], where the latter is very fast which makes it an excellent method for including statistical support when using large datasets. In recent years, the traditional Felsenstein bootstrap [44] has become replaced by the normalized bootstrap, which ponders the number of differences between local tree topologies [45]. All of these are established and well-known methods. Since this essentially concerns a superfamily that consists of several subfamilies, sequences from homologues need to be classified. We will use the recently developed HMMERCTTER [46] since it has been shown to outperform the Panther phylogenomics suite on a number of protein superfamilies, among which is the GH28 protein family.

The first type of character that is used to explain functional differences is that of specificity determining positions (SDPs) that can be detected by various approaches. The basics of most approaches is best known as evolutionary tracing [47,48], which uses a phylogenetic hierarchy and corresponding multiple sequence alignment (MSA) to identify positions that show conservation at a certain hierarchic level. Other methods such as SDPfox [49] use mutual information (MI) between MSA columns and clustering to identify Clustering Determining Positions (CDPs). Since selection of substitutions can occur as a result of drift rather than selection, these are not necessarily SDPs. Other MI methods such as Mistic [50] compare the columns of an MSA. High MI between two MSA columns suggests the corresponding positions somehow interacted during evolution. We combine SDPfox with Mistic based on the premise that CDPs that show high MI are unlikely to have evolved by drift. We apply this using a network approach and a network connectivity score to select the most probable SDPs.

The second type of character that can confer functional differences consists of subfamily specific inserts that typically form surface-exposed loop regions. These can also be detected by a number of means, of which structure modelling has become the most reliable since the recent advent of version 2 of Alphafold [39]. Alphafold not only outperformed all other structure model methods in the latest CASP, but as also shown, its reliability is close to that of actual structures [51]. A sequence-based method is the MEME suite, which identifies conserved subsequences.

In this study, we review the knowledge that has been gathered throughout the years and perform a systematic computational analysis to bring forward a comprehensive view on the evolution of the GH28 superfamily in fungi, directed at the identification of sequence traits that may be linked to certain functional specificities. We perform profound sequence mining using state of the art clustering and classification techniques. We use phylogeny to show the relationships of fungal enzymes that seem to have diversified to their full extent in ascomycetes only. We establish two rather than three different types of exoPG and show evidence for two types of endoPG as well as at least four exoRG subfamilies. The exoPGs have different additional loops that, according to Alphafold made structure models, result in more closed binding clefts. EndoRGs have diversified but this has not led to clear functional subfamilies. We also show that functional diversification in fungi has been independent from eventual diversification in bacteria and plants. Based on our systematic analyses and previously obtained knowledge, we propose a comprehensive nomenclature for particularly ascomycete GH28 enzymes.

## 2. Materials and Methods

### 2.1. Sequence Mining

Training. Pfam’s [52] hmmer profile PF00295 (*Glycoside hydrolase family 28*, [53]) was used to run hmmsearch in HMMER [54] site [55] vs. SwissProt database (version swissprot v.2019_09) [56], using the inclusion threshold. Searches were restricted to: Fungi (taxid: 4751); Insects (taxid: 50,557); Bacteria (taxid: 2); Archaea (taxid: 2157); Nematoda (taxid: 6231); and Viridiplantae (green plants, taxid: 33,090). No hits vs. Archaea and only one for Nematoda were detected, and since no nematode sequences were identified in the target sequence as described below, this was excluded from further analysis. The search against Fungi did not yield hits from basidiomycetes. Consequently, we searched for sequences from basidiomycetes. In short, we selected a single sequence from different orders/genera that we identified in a BLAST search [57] using AAF68403.1, an endopolygalacturonase sequence from *Chondrostereum purpureum*, against Reference Proteins basidiomycetes at NCBI. The nine sequences represent agaricales, boletales, russulales, polyporales, cantharellales, rhodotorulla, ustilago and exobasidiomycetes. Of these, two were partial and excluded. Single sequences from the species *Leptinotarsa decemlineata*; *Anoplophora glabripennis*; *Sitophilus oryzae*; and *Agrilus planipenni* were selected from a BLAST using AAF68403 against the insect reference protein database. Three endoPG sequences from *Phytophthora* species were selected to represent oomycetes. We selected 24 sequences from taxonomically representative plants from 1000 BLAST hits obtained using PGLR-VITVI as a query against green plants reference proteomes. Sequences for 40 PGs from taxonomically representative bacteria were identified from Pfam’s PG dataset [58]). Their full-length sequences were obtained from Uniprot [59] using their Uniprot codes. Then finally, we added all PG sequences for which a structure has been resolved except for the sequence of the 4MXN structure, since this is a partial sequence that interferes with HMMERCTTER clustering. The training sequences are in Appendix A.

Target. We used Pfam’s hmmer profile PF00295 (*Glycoside hydrolase family 28*) to run hmmsearch vs. Reference Proteomes in HMMER (Uniprot version: uniprotrefprot, v.2019_09), using the inclusion threshold. Independent datasets for Fungi, Oomycota (taxid: 4762), Insects, Plants and Bacteria were generated. Each dataset was aligned with MAFFT-G-INS-i (v7.453) [40]. Afterwards, a reference sequence with a resolved 3D structure (1CZF_A) was included in each MSA using MAFFT-add. Because of the large variability in the proteins’ ends, the resulting MSA was N- and C-end trimmed using the known secondary structure of 1CZF_A. For this, MSAs were visualized and trimmed using Aliview (v1.27) [60]. Finally, sequences in each dataset were automatically scrutinized using SEQrutinator. This pipeline objectively removes (i) very short sequences (with length < 65% of the reference), (ii) sequences with poorly represented inserts, thus instigating unique gaps in the MSA, (iii) sequences lacking subsequences present in most sequences, thus presenting long gapped regions in highly occupied columns, and (iv) outlier sequences based on their HMMER score when using a HMMER profile built with the dataset. Typically, i, ii and iii can identify and remove sequences with annotation errors (e.g., sequences with incorrect intron predictions), whereas iv can potentially identify false positives and pseudogenes. Hence, the automatic scrutiny yields a higher quality dataset, which can be utilized more reliably in further steps. Because of the iterative character of SEQrutinator steps, and the large number of sequences in the datasets, we used FAMSA [41] as the alignment method in SEQrutinator.

Hmmsearch was used to identify sequences with repeats, which obtain high total scores as a result of the repeats. Sequences with a hmmer score higher than any of the training sequences were checked for the presence of repetitions using Dotlet [61] and, upon confirmation, removed from the dataset.

### 2.2. Phylogenetic Reconstruction

For phylogenetic reconstruction, datasets were aligned using the MAFFT-G-INS-i method [40]. Afterwards, datasets were pruned using successively trimAl v1.2rev59 [62] with mode *gappyout* and BMGE v1.0 [63] with entropy threshold of 0.8. The resulting pruned MSA is used to reconstruct the phylogeny, using maximum likelihood methods provided by PhyML v3.3.3:3.3.20190909-1 [42] or FastTree v2.1.11 [43]. PhyML was performed with the Akaike Information Criterion. For statistical support, 1000 bootstrap were obtained with SeaView v4.0 [64], and the corresponding trees were generated with FastTree. Statistical support was analysed with transfer bootstrap expectation (TBE), calculated with Booster [45] using the PhyML-generated tree as a reference tree. Trees were visualized using Dendroscope [65] and/or iTOL [66].

### 2.3. HMMERCTTER Clustering and Classification

HMMERCTTER [46] clustering was performed by consistently accepting clusters that show 100% precision and recall, rather than searching for alternative smaller and possibly better clusters. HMMERCTTER classification was performed using the default procedure unless otherwise indicated.

### 2.4. MEME Motifs and Logos

Unaligned sequence sets for each subfamily were extracted from the final complete sequence set and, as such, were used to identify motifs using MEME [67] suite online [68], and using the classic mode setting to construct the logos, we extracted the corresponding subMSAs from the subfamilies and their counterparts (i.e., the remaining sequences). Gapped columns, defined by the reference sequence for each subfamily, were removed resulting in pseudo subMSAs. For SDP logos, a script was used to isolate SDP columns into pseudo SDP MSAs. Logos were made using weblogo [69].

### 2.5. Structure Modelling and Consurf Analysis

Structure models were obtained from ColabFold: AlphaFold2 [39] at [70]. Consurf [71] analysis was performed using the online service [72] with default settings. Structures and structure models were analysed and decorated using PyMOL [73].

### 2.6. Generation of Specificity Determining Networks (SDN)

MSAs for target datasets were trimmed, if required, based on the reference structure to maximally 5000 columns in Aliview [60], excluding N- and C-terminal parts of the MSA. The MSA was then used to identify CDPs using SDPfox’s sdplight module [49]. The same MSA was used to determine MI using Mistic [50]. With these inputs, we used a dedicated Python script to generate an SDN. The basic idea of the SDN is that highly interconnected CDPs are less likely to have resulted from drift and, as such, are more likely to be SDP. The script takes all CDPs determined by SDPfox with z-score > 3.29, and all MI z-scores > 3.29 according to Mistic. The Mutual Information Connectivity Score (MICS) of a CDP is a score that, for a certain MI threshold (MIT), expresses its total MI to other CDPs (the sum of its z-scores) divided by the number of connections the CDP’s node i has to other CDP-nodes, according to the MIT.
(1)$${\mathrm{MICS}}_{i}=\frac{\sum {\mathrm{MI}}_{i}}{\sqrt{N-1}}$$
where N = number of CDPs that have at least one connection with a z-score larger than the MIT.

The script determines a range of MICSs for all CDPs with MITs in between 3.29 and the highest MI z-score of the CDPs mutually. Note that the MICS of a node, as such, will increase until the corresponding CDP is excluded by the MIT.

Then, an SDN score or Specific Network Score SNS is calculated for each MIT with the formula
(2)SNS=MIT×∑MICS

By increasing the MIT, the number of nodes in the network will decrease, and the SNS will typically increase until a maximum is reached, followed by a decrease in the SNS. As a result, the SNS is a compound measure that weighs the number of connections to the strength of the connections. We, therefore, select the best SDN as that presenting the maximal SNS. Once the SDN is determined, the script ensembles 1000 random networks with the same node number as the SDN, using all nodes (both CDPs or not CDPs) and the same cut-offs. This implies that, by chance, random networks may or may not contain actual CDPs. Then, the SNS for each random network is calculated, as well as the z-score for the SDN in the distribution of the randomly generated networks.

## 3. Results

### 3.1. Phylogenetic Clustering of GH28 Is Explained by Taxonomy and Functional Diversification

Sequence mining was performed using a HMMERCTTER clustering and classification strategy, to ensure highly sensitive but, nevertheless, precise sequence mining. HMMERCTTER [46] clusters training sequence sets using their phylogeny and uses HMMER [54] profiles that represent MSAs of these clusters to classify additional homologues from large target sequence sets to each of the defined clusters. To obtain high accuracy and sensitivity, its clusters identify their own sequences with a HMMER score higher than the HMMER score of other sequences from the complete dataset. As such, the partition that results from the initial clustering shows 100% precision and recall (P&R) and this defines HMMER score thresholds to be used as specific inclusion cut-offs in an eventual subsequent classification of sequences from large datasets. The classification is iterative and dynamic by which sequence mining is also sensitive, whereas the 100% P&R rule maintains high specificity. To apply HMMERCTTER for sequence mining, we need a small but representative GH28 homologue sequence training set with high-quality sequences as well as a large target set with sequences from GH28 homologues.

The training sequence set has 255 sequences and was obtained as follows: We identified 164 sequences in the SwissProt database [56] using the Pfam [52] profile for GH28 [53]. These derive from 125 ascomycete, four mucoromycete, 30 plant and five bacterial PG homologues. We added seven sequences from seven taxonomically representative basidiomycetes, four sequences representing insect and three sequences representing oomycete PGs. Most plant sequences formed a single cluster in a preliminary tree but a single sequence, PGLR-VITVI, clustered with the five bacterial sequences, suggesting bias (i.e., lack of at least plant homologues). Hence, we extended both the plant and bacterial sequence sets adding 24 and 40 sequences, respectively. The identified sequences are in Appendix A.

The target set was obtained using HMMER [54] and SEQrutinator. Sequences from PG homologues were obtained using the general Pfam [52] profile and hmmsearch at the HMMER website [55], using Reference Proteomes. We found 3842 homologues in fungi, of which 2989 from ascomycetes, 733 from basidiomycetes, 88 from mucoromycetes and the remainder from other taxa. In plants, we identified 7098 homologues, whereas for insects a mere 45 sequences were retrieved. Oomycetes showed 191 instances, whereas a total of 3125 sequences were identified in Bacteria. The identified sequences are in Appendix A. Many complete proteome sequence sets contain partial sequences, pseudogene sequences and sequences that derive from incorrect gene models. These sequences hinder classification and provide noise in sequence analyses. To remove most sequences for these non-functional homologues, we performed automated sequence scrutiny using SEQrutinator. A total of 2445 sequences were removed (from Fungi: 443, Insects: 4, Oomycetes: 65, Bacteria: 158, Plants: 1775), resulting in a total of 11,856 sequences. In the first attempt to classify this set of GH28 homologues, we identified many conflicts that we could ascribe to sequences that have resulted from one or more duplication events. Hence, we scrutinized the target set for sequences with repeats, after which we remained with a target set of 11.789 sequences. The final set of target sequences is in Appendix A.

A preliminary ML tree (Figure 2) for the training set was obtained following sequence alignment and MSA trimming. All fungal GH28 sequences cluster in complex clade G1, with hierarchic subclustering of functional classes, whereas bacterial PGs cluster in clade G2. Plant PGs appear in two subclades, G1-4 and G2-1, hence in each of the two major clades. GH28, most likely, did evolve in plants, where its substrate is found, and the tree topology suggests that two different homologues gave rise to fungal homologues in clade 1 and the bacterial homologues in clade 2. Insect and oomycete endoPGs are surprisingly indistinguishable from fungal endoPGs, which may imply moderately recent gene transfer events as has been suggested earlier [16,74], combined with a similar functional constraint. The treefile is in Appendix A.

Besides the occurrence of taxonomic signals, the major tree topology clearly shows various signs of functional diversification. HMMERCTTER clustering of the training set was performed with the final objective of identifying clusters that are 100% P&R and correspond to a known subfunctionalization. As such, we performed inceptions, which consist of realigning the subset of sequences from a HMMERCTTER cluster, followed by phylogeny and again HMMERCTTER clustering. Hence, inception results are in a hierarchic clustering guided by the strict HMMERCTTER rules. G1-1 contains all known fungal endoPGs, as well as oomycete and insect endoPGs, whereas fungal endoRGs fall in G1-2 and canonical plant endoPGs in G1-4. G1-3 shows a complex clustering with four clades containing exoPGs (PGXA (including PGLRX coded sequences); PGXB; PGXC; and mucoromycete exoPGs) as well as an exoRG clade an endoXG clade.

Classification of the target sequence set was directed at studying fungal homologues. We used the partition that consisted of the entire G2 cluster, whereas the G1 cluster was represented by G1-1, G1-2 and G1-4 with G1-3 represented by six subclusters named as shown in Figure 2, representing mucoromycete exoPGs, PGXA, PGXB, PGXC, endoXG and exoRG. We joined all fungal sequences from the training set with those that were classified from the target set. Upon alignment and MSA trimming, we obtained an MSA with 331 columns which, given that the PDB file of 1CZF shows a mere 335 residues, indicates the MSA is high quality and provides sufficient information for reliable tree reconstruction. The ML tree of the fungal sequence set is shown in Figure 3A, showing the functional classes or subfamilies. Figure 3B shows the taxonomic distribution. Appendix A shows the consensus tree made with 1000 bootstraps, Appendix A contain the treefiles of the ML and consensus tree, respectively. Table 1 shows the number of subfamily homologues per class and/or division. Trees of the various subfamilies with ascomycete distribution at the taxonomic class level are gathered in Appendix A As expected, the largest subfamily is that of the endoPGs (34% against 25% exoPG). Surprisingly, there appears to be relatively many exoRGs, namely, 18% as compared to 15% endoRGs.

The tree topology differs slightly from that of the training tree. Particularly the endoXG clade appears as a direct sister-clade of endoRG, albeit with low bootstrap support (0.85, see normalized bootstrap consensus tree Appendix A). Furthermore, the PGXC clade is now more tightly related to the endoPG clade, supporting the idea that PGXC and PGXA/PGXB evolved independently. Although the PGXA and PGXB bifurcation has statistical support of 1, sequence analysis must be performed to see if A and B are likely also functionally different. EndoPG and, in particular, exoRG, do show clear subclades that may indicate further functional diversification. The endoRG clade is also large and many fungi have several paralogues, but the clade does not appear to show clear signs of further conserved subfunctionalization. Nevertheless, we made inceptions for all three of these large subfamilies (Appendix A). EndoXG is relatively small with typically only a few paralogues and was as PGXA, PGXB and PGXC not used for subfamily specific diversification. Appendix A shows extracts from Figure 3 with a more detailed taxonomy distribution for these subfamilies.

### 3.2. Only Ascomycetes Show the Full Extent of GH28 Diversification

Despite our efforts to perform sequence mining with an unbiased training set, the classification shows that particularly many basidiomycetes appear to have both exoRGs and endoRGs as well as exoPGs from the PGXC class, whereas these were represented in the training set by ascomycete homologues only. Most basidiomycete sequences appear in small clusters (Figure 3B and Appendix A, note these have different coloring), whereas some appear in clusters with mostly ascomycetes sequences. We also detected three PGXA and eight endoXG homologues but no PGXB homologues in basidiomycetes. This suggests that also these GH28 subfamilies did exist prior to the bifurcation of asco- and basidiomycetes, despite the low numbers of endoXG and PGXA homologues in basidiomycetes. Nevertheless, functional diversification only shows its full extent in ascomycetes. As such, we decided to study functional diversification using the ascomycete sequences only, to avoid putative signals from drift that may occur comparing sequences from distant taxa. Therefore, we selected the ascomycete sequences from which we removed sequences that lack D180 and/or both D201/[DE]202 since we consider these as non-functional and may instigate noise in the sequence conservation and diversification analyses. The final ascomycete sequence set is in Appendix A.

### 3.3. Functional Diversification in the Ascomycete GH28 Protein Family Can Be Explained by Subfamily Specific Loops That Cover the Binding Groove

We constructed a novel ascomycete dedicated tree and selected clades to ensure high conservation. In other words, we excluded sequences that appear to have evolved away from what seems like a functional class. The decorated tree showing bootstrap and clustering information is in Appendix A and Datafile S8.

We first analysed the structural basis of the different subfamilies. To do so, we selected the experimentally solved 3D models 1CZF as reference for endoPG, 1RMG for endoRG and 4C2L for endoXG. Due to the lack of available resolved structures for the other classes, we generated models for exoRG (A0A194WYS5_9HELO), PGXA (PGLRX_ASPFU), PGXB (Q7SAI8_NEUCR) and PGXC (A0A1L9N956_ASPTC) with AlphaFold [39]. We chose these sequences for modeling as they appear as the best representation for the subfamily (i.e., they have the highest HMMER score of all the sequences in a subfamily specific HMMER screening). The endoPG, endoRG and endoXG structures present an open binding groove (Figure 4), whereas all exomode subfamilies have one or more loops that appear to close or cover the binding groove. It seems that this is a general feature since bacterial endoPG (PDB identifier 1BHE [75]) and bacterial exoPG (PDB identifier 2UVE [76]) show similar characteristics as their fungal counterparts. Interestingly, the major covering loop of PGXC appears located in a different region than in the rest of the exomode classes. We compared the four models of the exoenzymes with the structure of 1CZF, using the basic structure description of Figure 1A. Appendix A shows the number of residues each reference enzyme has in its turns. All exoenzymes have inserts in T2-1 and T3-3, PGXA and exoRG have an insert in T1-10 whereas PGXC has an additional 25 amino acid long insert in T3-11. Appendix A shows how these inserts loop above the binding groove. Interestingly, all exoenzymes have an additional PB1-13, whereas the conservation of disulphide bridge SS4 shows that this corresponds to coil in the endoPG 1CZF, rather than being related to the fourteenth stack of endoRG 1RMG (see also Appendix A). It is also noteworthy that both α-helices and β-sheets are more exposed to the surface in endo-subfamilies.

The structural differences indicate that at least part of the functional differences occur as a result of additional subsequences. Next, we investigated if these subsequences can be identified by motif analyses. We performed the classic motif analysis provided by MEME suite [67]. Appendix A shows sequence logos of the three motifs that were identified for each subfamily, contrasted by the logo for the rest of the superfamily’s sequences. Appendix A shows, besides the aforementioned PB1-12 in exoenzymes and the additional stack of endoRG, the motif regions in the different structure models. Clearly, some of the motifs concern the site and subsites −1 and +1, whereas others correspond to loop regions that mostly contribute to more distant subsites. Most particular is that the most significant PGXC motif occurs in T3-11 (See Appendix A) which conceptually corresponds with subsites +2 and +3.

### 3.4. Evolutionary Conservation Analysis

Since the PG, RG and XGs have different substrates, we also expect differences in the binding groove, other than those provided by additional loops. Then, exomode enzymes are different in the extent that they do not require +2 or +3 subsites. We performed evolutionary conservation analysis using Consurf [71,77] to see if the different subfamilies show differences in the conservation of the binding groove. Consurf basically calculates a ranking of conservation using the hierarchic structure of a phylogenetic tree, similar to evolutionary trace [48]. As such, it will identify trends in conservation in a hierarchic manner.

We used datasets (MSAs and phylogenies (Appendix A)) dedicated to the seven major subfamilies in ascomycetes. All structures were aligned to 1CZF (chain B) and decorated according to their respective Consurf outputs. Figure 5 shows the groove regions corresponding to stack 2 to stack 5, stack 6 to stack 9 and stack 10 to stack 14. First, conservation is, as expected, high in the central region that includes catalytic stacks 7 and 8. Stack 9 also shows high conservation whereas stack 6 seems to be a bit less conserved. This suggests stack 7 to 9 rather than stacks 6 to 8 as the major contributor of subsites −1 and +1.

In the more N- and C-terminal regions that correspond with the more distant subsites, there is conservation but the patterns differ among the endomode subfamilies. This suggests substrate binding will occur at different positions for the different subfamilies. The clearest pattern that we identify among the endomode subfamilies is that endoRG shows less conservation in both the N-terminal and the C-terminal regions. This can be related to a high divergence of rhamnogalacturonan, while it has no clearly conserved subfamilies that would contribute to Consurf conservation signal. Interestingly, stack 14, which is only present in endoRG, does show moderate conservation. This may be explained by either a scenario in which endoRG has an additional subsite or by a scenario where the +2 subsite is a bit larger to bind a side moiety.

The exomode enzymes show interesting differences that corroborate PGXC as a rather different enzyme, suggesting that also PGXA and PGXB may be functionally different. ExoRG shows a conservation pattern that can be expected for an exomode enzyme: the C-terminal stacks are less conserved, in agreement with a lack of subsites beyond +1. PGXA still shows quite some conservation in particularly stacks 10 and 11. PGXB shows surprisingly low conservation in stacks 4 and 5. Perhaps, the most stunning result is that PGXC shows the opposite of the expected conservation pattern. It is highly conserved in stacks 10, 11 and 12, whereas it is less conserved in stacks 2, 3 and 4, which supposedly contain subsites −5 to −3, relevant for exomode enzymes.

### 3.5. Superfamily Specificity Determining Positions Show Large Physicochemical Differences in the Binding Groove

Specificity is not only determined by loops and other subsequences, it is often determined by a combination of SDPs in three-dimensional space. We used a combination of SDPfox [49] and Mistic [50] to identify SDPs and their mutual relations using network analysis and a Specific Network Score or SNS. A major difficulty in these analyses is the fact that superfamily evolution actually consists of the independent evolution of various subfamilies. This leads to overlapping signals that act as noise and can lead to incorrect conclusions. Hence, it is advisable to perform various comparisons. We performed a global analysis, which will have decent predictive power given the size of the dataset, as well as dedicated analysis, which will have decent predictive power given the lack of noise.

First, we identified networks of SDPs with all functional classes clustered individually, resulting in seven distinct clusters to evaluate the effect of overlapping signals or noise, we also performed SDP identification based on the three major phylogenetic groups (i.e., all-3 clusters: endoPG + endoRG vs. exoPG + PGXA + PGXB + endoXG vs. PGXC, Appendix A). Given that PGXC is an isolated group and we have shown it has rather different characteristics (Appendix A, Figure 5), we envisaged this could potentially introduce noise to the analysis. Together with the fact that it is the group with the smallest amount of information (with only 92 sequences), we choose to perform both the above-described analyses also excluding PGXC (i.e., 6 clusters-No PGXC and 2 clusters-No PGXC).

The analysis of all seven clusters resulted in an SDN with 36 SDPs (Figure 6). A total of 45 SDPs was identified when PGXC was left out of the major analysis (Appendix A). When larger clusters were considered, 18 SDPs were found (Appendix A), with 17 in the absence of PGXC (Appendix A). A total of 13 SDPs was identified in all four analyses (Appendix A). Network connectivity analysis determines a Mutual Information Connectivity Score (MICS) that weighs both the MI z-score of connections and the connectivity for each SDP. It then determines an optimal network for which the MI z-score is used as a cut-off. The SNS of the resulting Specificity Determining Network (SDN) has a z-score of 207.92 when compared in a distribution of 1000 randomly generated networks with otherwise similar characteristics. This indicates the network involves the more highly MI interconnected SDPs (Figure 6B). The logos corresponding to the SDPs in all sequences for each class reveal different patterns of subfamily specific conservation (Figure 6C). For example, SDP57 is very conserved in the PGXB subfamily (Ile90), but shows different degrees of variability in all other classes, although we typically observe hydrophobic residues in the homologous structures. SDP125 is very conserved in endoPGs (Lys125), variable in endoRGs (showing most commonly G or A), and with a substitution by R in all exoclasses and endoXG. SDP260 shows typically conserved bulky hydrophobic residues in exoclasses (e.g., Trp315 in PGXB) and endoXG (Tyr288), whereas it is variable in endoPG or not present at all in endoRG.

The SDPs are located in different segments of stacks 2 to 11 of the protein (Figure 6D). The majority of them, i.e., 25 of 36, are associated with turns adjacent to the binding groove, with five SDPs in T3-4, three in T3-5 and two SDPs in either T3-2, T3-3, T3-6, T1-9 or T1-11. When we restrict our analysis to the more solid set of 13 shared SDPs we find that eight of them (125, 150, 184, 200, 207, 232, 233 and 260) are associated with regions defining the binding groove. Furthermore, except for SDP184, the shared SDPs have a high MICS and form a subnetwork (Figure 6A). This result suggests an important degree of co-evolution among specific positions and supports the suggested role of the turns on the accommodation of substrate in the binding groove in ascomycete GH28 enzymes. The shared SDPs also show a high degree of conservation within the distinct subfamilies (Figure 6E) and different physicochemical characteristics in at least one of the subfamilies. For example, SDP125 is an amphipathic lysine in endoPG, a glycine in endoRG and a basic arginine in the other subfamilies. SDP260, which is located close to the binding groove, shows almost strict conservation of the bulky tryptophan in exomode classes while showing mostly the less bulky and more polar tyrosine in endoXG (Figure 6C,E). It is less conserved in endoPGs, being replaced by less bulky residues, and appears to be missing in endoRG. Similar observations can be made for other shared SDPs associated with the binding groove. Again, other shared SDPs with high conservation (57, 74, 78 and 105) are found in other regions of the protein.

To shed light on the diversification related to the rhamnogalacturonan substrate, we compared endoPGs in two independent analyses with endoRGs and exoRGs, respectively. We identified SDNs with 61 and 43 SDPs of which a total of 15 were shared (Figure 7 and Appendix A). Several of these commonly shared SDPs in endoRG and exoRG show a high degree of physicochemical conservation, as indicated in Figure 7. Moreover, some of the SDPs identified in both analyses present high MICS values, irrespective of whether as part of the complete networks (Appendix A) or as part of the sub-SDNs that include the shared SDPs only (Figure 7).

### 3.6. Specificity Determining Positions and Rhamnogalacturonan

Only three positions, 207, 333 and 150, according to 1CZF, have the same or a physicochemically similar residue in both exoRG and endoRG. N207K and Q150[FW] are two of four SDPs that are envisaged to be in close contact with the substrate, as demonstrated by their virtual vicinity to the galacturonate ligands of the aligned 1KCD structure. In the endoRG network, SDP207 (Lys203) interacts strongly with Arg128, which neighbors Phe151. We envisage these residues may be important for the diversification of site −1.

Interestingly, the fourth site SDP concerns His223, strictly conserved in endoPGs. This residue has been suggested as being required to create the correct physicochemical micro-environment of catalytic aspartates 201 and 202 [30]. Indeed, the SDP is not connected to the three SDPs supposedly involved in the diversification of site −1, but it is interacting directly with Lys196, being Gln200 in endoPG, a direct neighbor of catalytic Asp201. This suggests Lys196 may play a similar role in endoRG as His223 has in endoPG. This interaction is absent in exoRG, which shows no significant conservation at SDP223 (unconserved 252 in exoRG). EndoRG Lys196 corresponds with Gly228 in exoRG that takes part in the major triad of the exoRG network, connecting to Tyr140 and Arg152 that position T3-4 above Gly228 and the catalytic site. This likely explains the high conservation and high MI z-scores of this triad. A fourth SDP with high MICS connected to the major SDP triad is Asn237. This is located at the start of T1-8, bordering the catalytic site. Finally, the third conserved SDP is S333P; it is located at the end of T1-12, and is possibly important for the structure at subsite +1.

### 3.7. Specificity Determining Positions and Xylogalacturonan

We also analysed the case of endoXG, an endo-mode class that is phylogenetically associated with exomode classes. We performed two independent SDP identifications: a substrate focused comparison, endoXG vs. the combination of endoPG and endoRG, and a phylogenetic or mode focused comparison, endoXG vs. the combined sequences of exoRG, PGXA and PGXB (see Appendix A). In these analyses, we identified 32 and 31 SDPs (Appendix A), respectively. Six of these SDPs were identified in both analyses (Appendix A), of which only Lys281 and Gly284 are conserved across all endoXG sequences (Appendix A). All six, except Phe176, map close to the binding groove (Appendix A). As such, this suggests these positions may be implicated in substrate selectivity for endoXGs. Interestingly, major SDPs 321 and 329 concern a disulphide bridge that endoXG shares with PGXA, PGXB and exoRG. Hence, it is identified in the substrate focused comparison only. In 4C2L it positions in T3-10 next to Tyr288 and may, as such, contribute to the hypothesized diversification towards xylo- and/or rhamnogalacturonan substrate.

Unique SDPs identified in the substrate focused comparison of endoXG are more commonly found towards the C-end, whereas those from endoXG vs. exomode subfamilies are also found towards the N-end (Appendix A). Since only a few uniquely identified SDPs are associated with the binding groove, we next focused on the major SDPs from the comparison of endoXG vs. exoPG/RG in the expectation that this may shed light on how functional diversification of endoXG, potentially from an exoclass ancestor, may have occurred. Notably, Cys85, Ser115, Phe176, Ser236, Lys260, Ser261 appear to surround PB1 (Appendix A), the central pleated β-sheet of the binding groove. Cys85 forms a disulphide bridge with undetected C88, which suggests this concerns a structural constraint on the binding groove rather than a direct functional constraint such as for substrate specificity. Gly284 is located in the middle of PB1-9, hence subsite +1 and is as Ser261 part of the central network described above. Leu120 is located next to the SS bond formed by Cys85 and Cys88, thereby possibly providing support. The last major SDP from this specific network is Ala138, part of a distant loop. Although these data do not allow for a molecular explanation, we can envisage that this network of SDPs is important for either the structure or the dynamics of the binding groove and, as such, these may be crucial for either endomode action of endoXGs or to provide processivity. Full SDP logos are in Appendix A.

### 3.8. PGXA and PGXB SDPs Do Not Show Clear Signs of Functional Diversification

PGXA and B form a single monophyletic group and share many characteristics; however, they also show some differences in their loop regions. SDP identification resulted in a highly significant network with 31 SDPs of which several show significant physicochemical differences (Appendix A). It contains two pairs of SDPs with relatively high MI, which in both cases can be explained by a structural constraint since the residues of both pairs 104–106 and 248–250 interact physically. Furthermore, we identified two different subnetworks: a subnetwork with a few SDPs with relatively high MIs centered around SDP296 and a subnetwork of SDPs with relatively high MICS and low MIs, centered around SDP73 (Appendix A).

The SDP296 subnetwork includes pair 104–106 as well as SDPs 164, 192, 209 and 240. SDPs 164 and 192 can be envisaged to affect substrate binding, but both are only conserved in PGXB. The other SDPs are all distantly oriented away from the binding groove. Hence, we have no clear explanation for these SDPs. Cys73 is part of disulphide bond SS1, absent in PGXB, with its counterpart Cys50 undetected as SDP. It connects to the other high MI pair of 248–250, as well as to high MICS SDPs 142, 231, and 243. High MI pair 248–250, as well as SDP243, are in the vicinity of the site. The nature of both the Cys50–Cys75 and 248–250 pairs suggests that this subnetwork concerns a structural constraint rather than a direct interaction with the substrate. A last noteworthy SDP is P308, which makes the PGXA specific loop happen (Appendix A).

### 3.9. Ascomycetes endoPG Show Diversification of the T3 Loops

High numbers of subfamily paralogues facilitate further subfunctionalization, which is possible if there is a specific functional constraint. Particularly endoPG, exoRG, and also endoRG, appear as subfamilies of which most species have many paralogues (See Table 1). Furthermore, and as previously mentioned, both the endoPG and the exoRG subfamilies show topologies that suggest two or more functional subfamilies have evolved. Processive and non-processive enzymes have been described. Then, different endoPGs are also known to show different efficiency with methylated pectin and particularly rhamnogalacturonan forms a very amorphous substrate. Given that both conditions for functional diversification are met, we decided to study if functional diversification beyond the known descriptions can be detected, with a particular interest in substrate specificity. We applied HMMERCTTER to the endoPG, exoRG and also the endoRG subfamilies to see if conserved subclades can be detected. These were then further analysed.

The HMMERCTTER clustering of the endoPG subfamily identifies two subfamilies with mostly ascomycete sequences, hereby declared as endoPGA and endoPGB, and a small subfamily with all mucoromycete and chytridiomycete sequences. A few basidiomycete sequences are found in the two major clusters, whereas the rest is clustered into 66 additional clusters (Figure 8A). We studied an eventual subfunctionalization of endoPG by comparing the two major clusters using SDPfox and identified a total of 62 CDPs that appear rather dispersed over the structure. Mutual information network analysis of the identified CDPs selected 27 SDPs (Figure 8B). We mapped the SDPs in 1CZF and 1HG8 as references for endoPGA and endoPGB, respectively. Six of the seven highest-scoring SDPs locate on one of the T3 loops that cover the N-terminal part of the groove, conceptually corresponding to sites −1 to −3 or −4 (Figure 8C). From the remaining SDPs two, Ala205 and Val286, map to PB1 on the C-terminal side of the enzyme, some other SDPs can be explained as possible compensation for the aforementioned SDPs, but no other SDPs that appear as possibly in contact with substrate were identified (Figure 8D, 205 is Alanine in both 1CZF and 1HG8). Figure 8E contains the SDP logos for the endoPGA and endoPGB subfamilies, showing overall considerable physicochemical differences. We suggest the common denominator X in its meaning of “unknown” or “unclassified” for all endoPGs that are not part of either the endoPGA or endoPGB subclasses.

Processive endoPG 1NHC and non-processive 1CZF are both part of endoPGA, the larger of the two endoPG subfamilies (Figure 8A). We mapped all endoPG sequences that have arginine at SDP91 (hitherto referred to as R91-sequences) on the phylogeny of Figure 8A. All R91-sequences, except for one, fall in the same sub-subclade of the endoPGA subclade. Since, in addition, most of the sub-subclades’ sequences are R91-sequences (Figure 8A), it concerns a conserved trait, specific for eurotiomycetes. Interestingly, the 1CZF homologue appears very close to the subclade with R91-sequences. We compared all sequences from the R91-sequence subclade with all other sequences from the endoPGA clade. From a total of 33 CDPs, cross-analysis of SDPfox and MI suggested 28 are actual SDPs (Appendix A). The network shows S91 does take part of the SDN but also that it has a low MICS. It has only a few connections, none of which with a high MI z-score but all nodes locate in the same N-terminal part of the groove (Appendix A). A second subnetwork centered around SDP135 also concerns a number of positions in the same part of the groove. Combined, they appear to enclose subsites −2 to −5. SDP logos are in Appendix A). Note that Appendix A shows the structure of an endoPGA from *A. aculeatus* (PDB identifier 1IA5 [78]), since 1CZF is very closely related to processive 1NHC.

### 3.10. exoRGs Show Diversification of Site and Close by Subsites

The exoRG clusters in four well-separated groups that are 100% P&R (Figure 9A), and we analysed exoRG diversification using a single SDPfox analysis in which we compared the three larger subfamilies, the fourth consisting of too few, a mere 10, sequences. Out of 77 CDPs, 47 were selected as probable SDPs by network connectivity analysis (Figure 9B). SDP logos show the diversification over the four subfamilies. Next, we identified which of these SDPs may be involved in site and subsite diversification. SDPs Asn148, Met176, Val205, Ala210 and Val252 are likely in direct contact with the substrate or catalytic residues, and we consider them active site SDPs (Figure 9C). Ser203, Thr204, Thr282, His284 and Asn301 surround these active site SDPs and have probably diversified to compensate eventual drawbacks of the catalytic SDPs. Met176 has the second-highest MICS being connected to all other catalytic SDPs and most of the other SDPs. Its orientation and that of the ligands from the aligned 1KCD suggest it corresponds to subsite −1 and, as such, to the rhamnose moiety, rather than +1 and the to be digested galacturonate. Ser231 and Phe248 are the counterparts of central disulphide bridge SS2 that holds the catalytic site together in most GH28 enzymes, including three of four exoRG subfamilies (Figure 9C,D).

Gly107 has the highest MICS and is together with Turn T3-3 also involved in subsite −1 or −2. SDP Trp92 is the last residue of PB3-3. T3-3 starts here in exoRGs with a short additional helix (Figure 9C). Then from SDP Met104 until SDP Ala115 it forms an extended loop that folds over the groove, placing SDP107 directly above the substrate and SDPs Thr176, Thr178, Ile181 and His183 of stack 6. Next to these on stack 7 are Ala210, a neighbor of what would be Asp183 in 1CZF, and Ile213 and Tyr214. Hence, many of the exoRG SDPs can be explained as taking part in the site that corresponds to the rhamnose in subsite −1 of the enzyme. The variation of rhamnogalacturonan is correspondingly found in the side chains of the rhamnose moieties.

### 3.11. A Small but Significantly Different endoRG Subfamily Is Detected

Although most fungi contain various endoRGs, HMMERCTTER clustering of the incepted dataset did not show clearly separated clades with many sequences (Appendix A). This does not mean functional diversification has not occurred but rather that it has not been conserved, by which we choose not to investigate it. Clades are small and typically with taxonomic bias and, given the complex topology, multiple comparisons will be required which makes it impossible to make predictions with significant reliability. However, while checking the dedicated endoRG MSA, we did identify a subfamily that has an insert, as compared to the rest of the subfamily. It has sequences from the four major orders of ascomycetes. Furthermore, note that the subsequences that correspond with the insert are not included in the trimmed MSA we used for phylogenetic reconstruction. Appendix A shows an alignment of a 3D model using the sequence that is closest to the common ancestor (i.e., it has the highest HMMER score) with the 3D structure of the endoRG from *A. aculeatus* mentioned earlier (PDB identifier 1RMG). Interestingly, the insert in T3-3 appears to have resulted in a loop covering the binding groove, similar to the loops that cover the binding groove in exomode enzymes.

## 4. Discussion

In this work, we evaluate the fungal GH28 protein family by means of computational analysis of protein sequences. On the one hand, it builds on existing knowledge; on the other hand, it forwards a number of hypotheses. These hypotheses are based on evidence provided by phylogenetic reconstruction and classification into six classes or subfamilies, namely, endoPG, endoRG, endoXG, PGXC, exoRG and finally PGXAB, which combines the former classes of PGXA and PGXB. We show evidence for further subfunctionalization of the exoRG into exoRGA, exoRGB, exRGC and the minor exoRGD subfamilies. For endoPG we suggest the subfamilies of endoPGA and endoPGB, while endoPGs that cluster elsewhere can be named as endoPGX. Classes or families and subclasses or subfamilies were then studied by a number of independent computational analyses. These analyses are relevant, due to a large number of sequences and were directed at the identification of patterns of sequence diversification at the superfamily level, a diversification that goes combined with sequence conservation at the family or subfamily level. In order to obtain the highest signal, we removed dubious sequences and restricted certain datasets to conserved clusters. It must be stressed that conservation forms strong evidence for function, but also that conservation by itself cannot identify the nature of the related functional diversification. The latter has various aspects such as pH and temperature optimality. We focused our efforts on explaining the classification in terms of substrate specificity and differences between the exo-, endo- and processive modes.

### 4.1. Functional Diversification in Fungi Has Resulted in Six Major GH28 Subfamilies

Functional diversification results from a combination of mutation and differences in functional constraint. Gene duplication results in functional redundancy, which is often related to the birth and death model of evolution [79]. However, functional redundancy allows for high sequence diversification that, whenever there is sufficient functional constraint, can result in functional diversification rather than the loss of paralogues. As such functional redundancy and functional diversification are related and have had their impact on many superfamilies, among which is the pectate-lyase-like gene family. Given the origin of the substrate polygalacturonan, the first steps have most likely occurred in plants, resulting in, among others, the pectate lyase gene family and the glycoside hydrolase family 28 (GH28). These have then independently spread to, among others, bacteria and fungi. Functional diversification of the GH28 family, related to different substrates such as homogalacturonan, rhamnogalacturonan and xylogalacturonan is most obvious in fungi—the subject of this study.

The phylogenetic trees from Figure 2 and Figure 3 indicate a number of aspects but we must stress that one must be careful drawing conclusions, since superfamily trees result from various forces. Where species trees are reconstructed under the premise of drift and negative selection, superfamily trees result from drift combined with strong functional constraints that are either negative or positive. The latter complicates things further since we do not know when a novel functional subfamily evolved, hence, we do not know when certain functional constraints came into effect.

The topology of the general tree in Figure 2 suggests that fungal GH28 enzymes have a single origin, as do bacterial GH28 enzymes. As such, not only bacteria and fungi, but also, most likely, plants, appear to have convergently evolved exomode enzymes. Comparing structures shows that exomode requires additional loops in the turns in between subsequent β-sheets of PB1 and PB3 (Appendix A). Consensus has it that there is little constraint against insertions in surface-exposed regions of proteins, which may explain for the proposed convergent evolution.

The topology of the fungal GH28 tree in Figure 3 shows four major clades. Besides the well-separated endoPG clade and the combined endoRG/endoXG clade, there is a third clade that harbors exoRG and exoPG class PGXAB, whereas the fourth clade contains exoPG class PGXC. The topologies of the trees in Figure 2 and Figure 3 differ slightly, putting both PGXC and endoXG with the other exoPGs. In a third tree that contains ascomycete sequences only (Appendix A), PGXC remains separated, whereas endoXG returns to the clade with exoRG and PGXAB. This tree has less taxonomic signal and may as such be the tree that most reliably shows the order of events. A likely scenario is that the exomode evolved as a result of loop regions in Turns 2-1 and 3-3, in a single or in multiple events. PGXC further evolved with the additional insert in Turn 3-11, whereas PGXA and exoRG obtained another insert in T1-10. In that scenario exoRG evolved from the same common ancestor as PGXAB, whereas PGXC evolved in an earlier event. Note that endoXG is processive [9] and, as such, also has exomode activity. We can envisage it has also evolved from the same ancestor as PGXAB and exoRG, rather than that is shares a common ancestor with endoRG. In another scenario, exoPG evolved only once. The plant and bacterial sequences do serve as outliers in Figure 2, in which PGXC clusters with the other exomode enzymes.

One of the major issues when investigating protein superfamilies is to pinpoint when a certain clade or cluster should be considered as a functional subfamily. Since conservation is related to function, we work under the premise that subfamilies that substantially differ from other subfamilies from the same superfamily show signs of conserved differentiation. Hence, sequences from a functional subfamily are more conserved among each other than when compared to sequences from other subfamilies. We used this idea to develop HMMERCTTER—a method and software for the clustering and subsequent classification of protein superfamily sequences. HMMERCTTER clusters superfamily sequences using a phylogenetic clustering and HMMER searches. The particularity of the HMMER searches is that each subfamily detects its member sequences with hmmer scores higher than that of other (superfamily) sequences, which is referred to as the 100% Precision and Recall (100% P&R) principle. This has been applied successfully in the clustering of both the α-crystallin domain protein family in plants [46,80] and the phospholipase C protein family [46], two paradigms in terms of existing functional classification. Here we applied it successfully to GH28. Note that the 100% P&R rule simply means that members of a certain cluster can be clustered with high fidelity. Based on the high fidelity, a researcher has to make the decision as to whether this cluster is also functionally different. Hence, it can help in functional clustering but additional evidence is required.

PGXA and PGXB form a nice example of the dilemma that one faces when elucidating a functional classification. The clades are monophyletic and are 100% P&R whether separated or when combined. Hence, additional evidence must be presented to claim these form two different functional subfamilies. In this case, the major difference between PGXA and PGXB consists of the additional insert of PGXA in T1-10 (Appendix A). Biochemical work has shown that PGXA has a higher specific activity towards xylogalacturonan than towards homogalacturonan [8]. However, the same study showed PGXC to have even better activity towards xylogalacturonan. Since, to the best of our knowledge, this has only been shown for single enzymes of the PGXA, PGXB and PGXC class, we find this is insufficient biochemical evidence for claiming PGXA and PGXB as different subfamilies. On the other hand, the taxonomic distribution is rather broad, which suggests both subfamilies contribute to survival and reproduction. In addition, we did identify an SDN with SDPs with clear physicochemical differences. However, since we cannot explain these SDPs in terms of substrate specificity, we conclude that there is also insufficient biocomputational evidence for PGXA and PGXB having different functions. It is however entirely possible these reflect enzymes that differ in, for instance, pH optimality. We restricted our analysis to possible differences in substrate and, hence the binding groove of the enzyme. Other instances of subclades that are 100% P&R were found for the endoPG and are particularly clear for the exoRG subfamily (Figure 8A and Figure 9A). We find the evidence for functional diversification among the endoPG and among the exoRG subclasses more convincing than the evidence for PGXA and PGXB.

### 4.2. Many but Not All Identified Specificity Determining Positions Correspond to the Site or a Subsite

SDPs are defined by biochemistry: they simply relate to functional differences. In this study, we applied a combination of computational methods to identify SDPs. Although the underlying methods are statistically sound, this does not mean that all positions identified are in fact SDPs. The network connectivity analysis can only show if the identified combination of SDPs is among the networks that show the highest levels of mutual information, which we accept as evidence for coevolution. We must understand, however, that there will always be a network with the highest z-score. On the other hand, the idea that a method will be able to identify all SDPs is a fixed idea.

In the global analysis, we identified 36 SDPs of which 25 associate with either a T1 or a T3 turn, dispersed over the entire binding cleft. Since many of these SDPs show high conservation in at least one of the subfamilies, these are very likely involved in substrate binding. The additional analyses, using three rather than seven clusters and those in which we excluded PGXC, were helpful in the sense that most SDPs were shared (note that 17 SDPs were identified in the 2 cluster analysis (Appendix A)). How underlying substitutions affect binding should be evaluated further, and goes beyond the scope of this study, which has a more general explorative character. Additionally, in most of the dedicated analyses, SDNs with SDPs near either the site or one of the subsites were identified. Note that many of the identified SDPs are not described textually nor are all presented in one of the many cartoons. On the one hand, a comprehensive description is not always the clearest description. In other words, we selected SDPs for which we could find a feasible explanation, albeit that we always focus on the SDPs with higher MICS and physicochemical differences in at least one of the subfamilies under investigation. Then, many SDPs can likely be explained as compensatory, as we did indicate for the case of the exoRG subfunctionalization. In this particular case with 47 SDPs, still many SDPs lack an explanation. In many cases, these SDPs may have resulted from some kind of structural compensation, which includes constraints from molecular dynamics. Others may be unimportant or are so-called background mutations [81] that have been selected by positive selection of other sites, as was hypothesized by Wagner [82].

As expected, we did identify His223 in the endoPG vs. endoRG and the endoPG vs. exoRG comparisons. In the case of the endoRG, it corresponds with Gly220, which is related to Lys196, which neighbors catalytic Asp197 and may play a role similar to that of endoPG His223 in creating the optimal microenvironment for catalysis as was suggested by van Santen et al. [30]. In exoRGs, an alternative for this electrostatic interaction was not found. Instead, the major subnetwork concerns a triad with Tyr140 and Arg152 that put T3-4 above Gly228 and the catalytic site. We cannot explain this in terms of creating a correct microenvironment, but can envisage this somehow affects catalysis or, alternatively, substrate binding.

Additionally, in the case of endoXG, we identified SDPs near the binding groove, particularly when focusing on the SDPs that were identified in both the substrate and the mode focused comparisons. Furthermore, we identified some SDPs that will have a structural impact in the same SDN as a number of SDPs that surround PB1 (Appendix A). Although this would make sense if endoXG has evolved from exomode enzymes, we stress this is merely a hypothesis. In addition, we cannot explain the mode switch in molecular detail. It should be noted that at least XghA (PDB identifier 4C2L) is a processive enzyme [9], hence with both endo- and exo-activities. More dedicated studies will need to be performed to shed light on both the origin and the exact mode of endoXG activity. Interestingly, for processive endoPGA, we also found SDPs surrounding PB1.

The study of PGXA and PGXB is rather different in the sense that both subfamilies are expected to have the same function, or in other words, we are not aware of known functional differences. SDP analysis did identify a large network of physicochemically varying SDPs, however, only SDPs 164 and 192 are close to the possible substrate and appear as conserved in PGXB only. This may indicate PGXB is more specific than PGXA, but confirmation using either wet lab or molecular docking will be required. Molecular docking may seem feasible due to the high quality of Alphafold modelling. However, it must be stressed that exomode enzymes likely depend on large loop replacements as we discuss below, hence this will also be a major research effort. As such, we stick to the idea that PGXA and PGXB are more likely a single subfamily.

Perhaps, the clearest result we obtained with SDPs is that for the hypothesized endoPG diversification. Six of the seven highest-scoring SDPs locate on one of the T3 loops that cover the N-terminal part of the groove, which suggests they may affect substrate specificity, the fifth scoring SDP, Asn207 locates opposite these loops on the other side of PB1. As such, this analysis has clearly identified SDPs and we sought the explanation in substrate specificity, hence, in methylation and or acetylation of homogalacturonan. The effect of methylation and acetylation has been studied for 1CZF and 1NHC (endoPGA) as well as for endoPGB 1HG8 [83]. Of these three enzymes, 1NHC was previously shown as the most versatile [84], and docking studies suggested Ser191_1NHC_ (Gly185_1CZF_ and Ser196_1HG8_) and Asp240_1NHC_ (Ser234_1CZF_ and Ser245_1HG8_) as crucial to allow binding of methylated polygalacturonan, whereas Ser91_1CZF_ and Val89_1HG8_ prevent the binding, as compared to Arg96_1NHC_. We did not detect any of these positions. However, 1CZF and 1NHC are not only both endoPGA enzymes, but they are also very much related and as was stated before, both also prefer pectate, whereas the *A. niger* endoPGB isoforms named C and E prefer pectin with low degrees of methylation [12]. Hence, either the clustering and observed conserved differentiation (Figure 8) relates to another functional diversification or it is related to another pattern of methylation and/or acetylation than that investigated by docking [83]. Clearly, any attempt to study the effect of methylation and acetylation is compromised by the complexity of these homogalacturonan decorations. This also demonstrates the distinct merits of having different types of study. The docking data [83] are solid since they come with previous biochemical analysis [84]. However, it concerns the analysis of three single sequences and enzymes and the results cannot be generalized. Our approach may be less solid but will, in all likelihood, provide data that are applicable in a more general way. An important aspect here is the role of epistasis and the background in which a certain substitution occurs. 1CZF and 1NHC form a paradigm that points towards this role, since they are very closely related but, nevertheless, rather different.

Indeed, 1CZF and 1NHC were central in the analysis of processiveness. The close relation of these enzymes explains, on the one hand, the rather easy conversion of R96S or S91R as described by Pouderoyen et al. [10]. On the other hand, it explains why S91 was not identified as the major contributor. The endoPGII of 1CZF has a number of prerequisite substitutions. Hence, the clustering we selected for this analysis is likely not the best one. In addition, it is known that processivity is ascribed to a number of endoPGB enzymes, such as BcPG3 [25] as well.

The exoRG family also has clear subfamilies and SDP analysis identified a number of SDPs at subsite −1, where the rhamnose with its various decorations is expected to bind. Additionally, here molecular docking must be performed to corroborate the hypothesis that the clustering and the SDPs are related to substrate specificity. A clear pattern of conserved differentiation was not found for endoRG, whereas this is clearly related to the same substrate. We have two possible explanations for this. First, the function of endomode enzymes is to degrade cell walls, whereas exomode enzymes are more related to nutrient provision. There is no need for various specific endoRGs that can digest specific conformations of rhamnogalacturonan; as long as it can hydrolyze the compound into smaller fractions, it has complied with its function. This constraint is stronger for exoRGs. Second, the endoRG subfamily is still rather large in the sense that it has many paralogues in many species. When compared to endoPG and exoRG, it appears however less structured in the sense that there are no concrete subfamilies. Hence, there is high sequence variation that has not resulted in conserved subfamilies, which is reflected by the low conservation of endoRG in the PB1 region (Figure 5). As such, most species have a set of highly variable endoRGs that provide adaptability in concert.

### 4.3. Loops That Cover the Binding Groove Determine Exomode Activity

There is no doubt in classifying GH28 sequences to one of the endomode classes. The comparative analyses we performed clearly indicate a number of differences. The loops, particularly in the T3 and the T1 turns are among those that were expected, based on literature [33,85,86]. It has also been suggested that endoRGs have more space in the active site to bind more complex substrates based on differences in loop structures [87]. Even although, in many cases, the actual evidence is scarce or exemplary, the structure comparisons that we made, clearly show both the T3 and T1 loops can play important roles. Indeed, we did find many SDPs, particularly on T3 loops.

To understand the role the turns and loops play in binding the substrate we must look at how the groove has been described. The structure of endoRG was described as “a large groove with walls defined by loop regions” [29], whereas endoPG 1CZF was described as an enzyme with a cleft that “is open on both sides, in accordance with the endohydrolytic character of the enzyme” [30]. The description for bacterial endoPG was, however, “10 turn righthanded parallel β-helix domain with two loop regions forming a “tunnel like” substrate-binding cleft” [75]. Bacterial exoPG 2UVE was suggested to have “a closed pocket that restricts the enzyme to the exclusive attack of the non-reducing end of oligogalacturonide substrates” [76]. Looking at the structures (Figure 4), we consider endomode enzymes to have an open groove rather than a tunnel, whereas exomode enzymes have loops that form a tunnel rather than a pocket. The loops that result from inserts in mostly T1 and T3 turns provide additional H-bonds that will be required to compensate for the loss of substrate binding energy as a result of the loss of subsites +2 and +3, as compared to endomode enzymes. Close inspection of the models for all exomode enzymes that we obtained using Alphafold teaches that, indeed, rather than a real pocket, there is a rather narrow tunnel. This does not exclude that the enzymes actually provide a pocket but, taking into consideration protein dynamics, a tunnel should be considered. Moreover, some strong additional evidence is provided by the processive endoPGA enzymes. The SDPs that we identified to be involved in the processiveness of endoPGA, occupy the N-terminal subsites, mostly flanking the groove. Provided that a processive degradation starts in endomode, it has to form an open groove. Upon the first digestion, the substrate is somehow relocated by one subsite before switching to its exomode. Since the difference in the endoPGA processive enzymes is determined in the N-terminal subsites, it seems rather unlikely that an N terminal pocket as might be formed for PGXA, PGXB and exoRG is required for exomode activity. PGXC might fit the pocket model better were it not for the conservation pattern we showed using the Consurf analysis (Figure 5). This actually points to a flaw in the pocket model since, for bacterial as well as for PGXA, PGXB and exoRG, the pocket model would require an inverted substrate orientation. Additionally, kinetically a pocket is not the preferred structure for exomode hydrolysis. Irrespective of the question of whether exomode hydrolysis is processive or not, in the pocket model the hydrolyzed galacturonate would be trapped in the pocket, prohibiting the binding of another fresh galacturonan substrate. Our analyses of exoRG diversification also suggest the subsite covered by the loops are the main factor in the diversification of exoRG, pinpointing the consensed orientation of the substrate. We stress this is an alternative model for exomode activity and that structure of exoPGs co-crystallized with substrate and/or molecular dynamics will be required to provide more evidence.

A novel subfamily that has evolved from endoRG in ascomycetes was detected. It has an additional loop above the binding groove that, in the light of our hypothesis regarding exomode activity, may be an example of convergent evolution of exoRG activity. Wet lab experiments will be required to prove or disprove that hypothesis. Another, perhaps, far-fetched hypothesis is that PGXC is an exoPG with inverted substrate variation. This is a mere parsimonious hypothesis, such that it can explain both the conservation pattern shown by Consurf analysis in Figure 5, the global structure analysis in Figure 4 as well as motif analysis shown in Appendix A. It also explains the odd behaviour of the PGXC clade in phylogenetic reconstruction. Although 12 SDPs for PGXC have been identified as part of the global SDP identification (Appendix A), we did not attempt to explain them. The combination of the fact that we cannot pinpoint its ancestor in a correct topology and that it has a number of strange characteristics (the high conservation of subsites +2 and +3 (Figure 5) and the large insert in T3-11(Appendix A)) makes this case difficult with too many “ifs”. It is however the most general exoPG, since only a few PGXAB sequences were found in basidiomycetes.

## 5. Concluding Remarks and Future Prospects

We have systematically evaluated sequences of the fungal GH28 protein family and provided its hierarchic classification. We explored the evolution of the protein family with a particular interest in substrate specificity and mode of action. We put forward a number of hypotheses that are based on solid sequence conservation and differentiation analyses as well as on structure modelling with Alphafold. We provided a rationale for the GH28 classification in terms of sequence- and structure-function relationships. We identified Lys196 in endoRG (numbering 1RMG) as probably functionally equivalent to His223 in PGs and XGs. An equivalent was not found in exoRGs. Processivity was further explained for processive endoPGAs, but not endoPGBs, due to a lack of conservation. We related the subdivision of ascomycete endoPGs to methylation and/or acetylation. The hypotheses that we propose are the result of an exploration of the relationship between sequence and structure and will require further investigation. The latter is best exemplified by the endoPG subdivision. Comparative sequence analysis, as such, typically identifies conserved traits. Once identified, a trait (here the suggested methylation/acetylation substrate specificity) needs to be corroborated. Indeed, proving or disproving certain hypotheses will require wet lab experiments. Another challenge that we have tackled successfully is that we propose an improved explanation for exomode activity. Rather than a pocket, as was suggested for bacterial exoPGs, we propose that loops form a tunnel and provide additional subsites to compensate for the loss of subsite +2 and +3, as compared to endomode enzymes. Additionally, the suggestion that a novel exoRG subfamily has evolved from an ancestral endoRG clearly requires confirmation. In many cases, however, molecular docking and/or molecular dynamics will be an important and effective tool. It will also require more actual structure data. In particular, we lack structures with long and differently decorated substrate analogues. A major obstacle is the diversity of pectin as a substrate. Enzyme-ligand structures with combined docking and biochemical experiments with well and less defined substrates will, in our opinion, provide more in-depth insights into the peculiarities of this interesting protein family. Docking may also provide insight into the substrate interaction of PGXC. Molecular dynamics will be required to shed light on how the various T1 and T3 loops invoke exomode and processive activity. It will be essential that docking studies will be accompanied by biochemical experiments by which good predictive models for substrate binding can be made, similar to what has been demonstrated, for instance, for truncated haemoglobins [88]. Hence, we foresee future breakthroughs in postgenome research to depend on structural biology, which investigates the relationship between sequence and structure for which the research we present here is only the first step.

## Figures and Tables

**Figure 1 jof-08-00217-f001:**
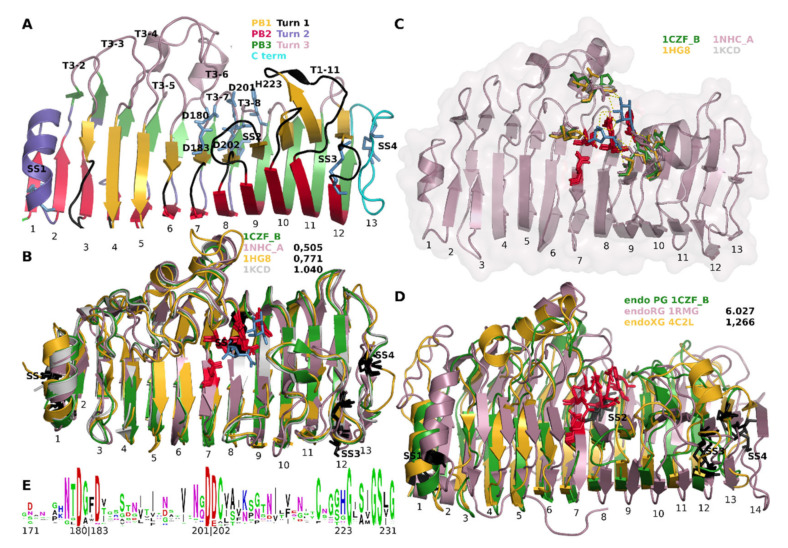
Global structure-function analysis of the Glycosyde Hydrolase 28 family. Numbers below the cartoons indicate the number of the separate helices or stacks. (**A**) Global architecture. Shown is chain B of 1CZF, endoPGI from *A. niger*. Gold: PB1 (stacks 3 to 12); red: PB2 (Stacks 2 to 12); green PB3 (Stacks 1 to 12). Black: Turn/Loop 1; Rose: Turn/Loop 2; Purple: Turn/Loop 3. The N-terminal part of the structure is supported by a single disulphide bridge (SS1), two disulphide bridges (SS3 and 4) support the C terminal part of the structure, one in Loop 1–12 and a second in the thirteenth and last stack. A fourth disulphide bridge (SS2) in Turn 3-7 supports the catalytic site that is located approximately in the centre of the slide (Turns 3-6 to 3-8). Disulphide bridges and catalytic residues are in black and blue sticks, respectively. (**B**,**C**) Structural alignment of endoPGs 1CZF, 1NHC, 1HG8 and 1KCD. RMSDs in Å indicate the structural dissimilarity with reference 1CZF. In blue stick are the two galacturonates co-crystallized with 1KCD. D180, D183, D200, D201 and H223 and corresponding residues in other structures are in red stick; (**C**) H bonding of galacturonates to N91, Q120, H150, D153, S201, R226, K228 and Y262 (1CZF) and corresponding residues (stick with colour according to the structure). (**D**) Structural alignment of endoPG 1CZF, endoRG 1RMG and endoXG 4C2L. Details as in (**B**); (**E**) Sequence conservation logo of GH28 superfamily region 171–231. Indicated are the highly conserved aspartates and H223, numbers according to 1CZF.

**Figure 2 jof-08-00217-f002:**
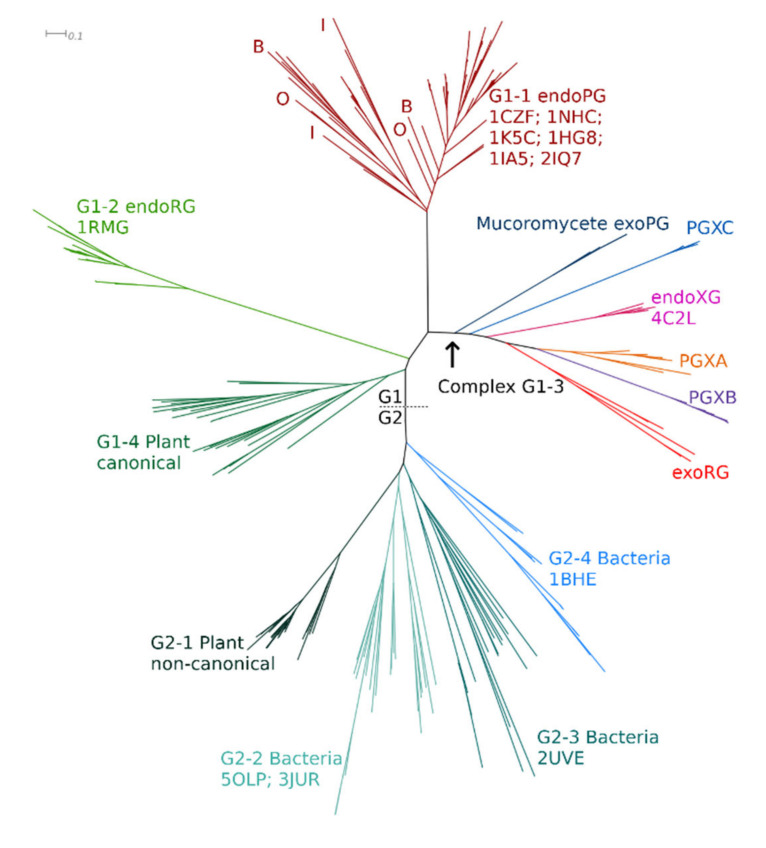
Phylogenetic relationship and functional/taxonomic clustering of GH28 training-set sequences. Sequences cluster with 100% P&R in major clusters G1 and G2, which were subclustered hierarchically to find functional clusters. Different clusters have different colors, are 100% P&R and represent either a functional class or a taxon. B: basidiomycete; O: oomycete; and I: insect (approximate indications), ascomycete homologues are not indicated. Sequences with resolved structures are indicated by their PDB identifier. The scale bar indicates the number of amino acid substitutions per site. The tree was constructed using the LG model, four fixed rate classes and a gamma shape parameter of 1.910.

**Figure 3 jof-08-00217-f003:**
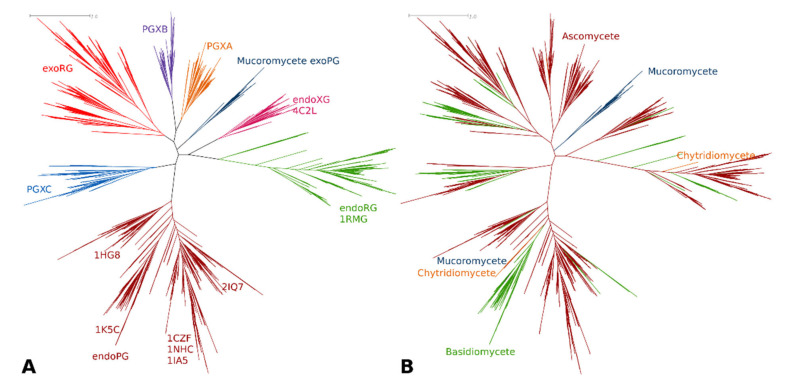
Phylogenetic relationship, functional clustering and taxonomic distribution of fungal GH28 enzymes. (**A**) Colors of clusters as in Figure 2. Sequences with resolved structures are indicated by their PDB identifier; (**B**) Taxonomic distribution of GH28 enzymes. For details per subfamily see Table 1 and Appendix A. The scale bar indicates the number of amino acid substitutions per site. The tree was constructed using the LG model, four fixed rate classes and a gamma shape parameter of 1.222.

**Figure 4 jof-08-00217-f004:**
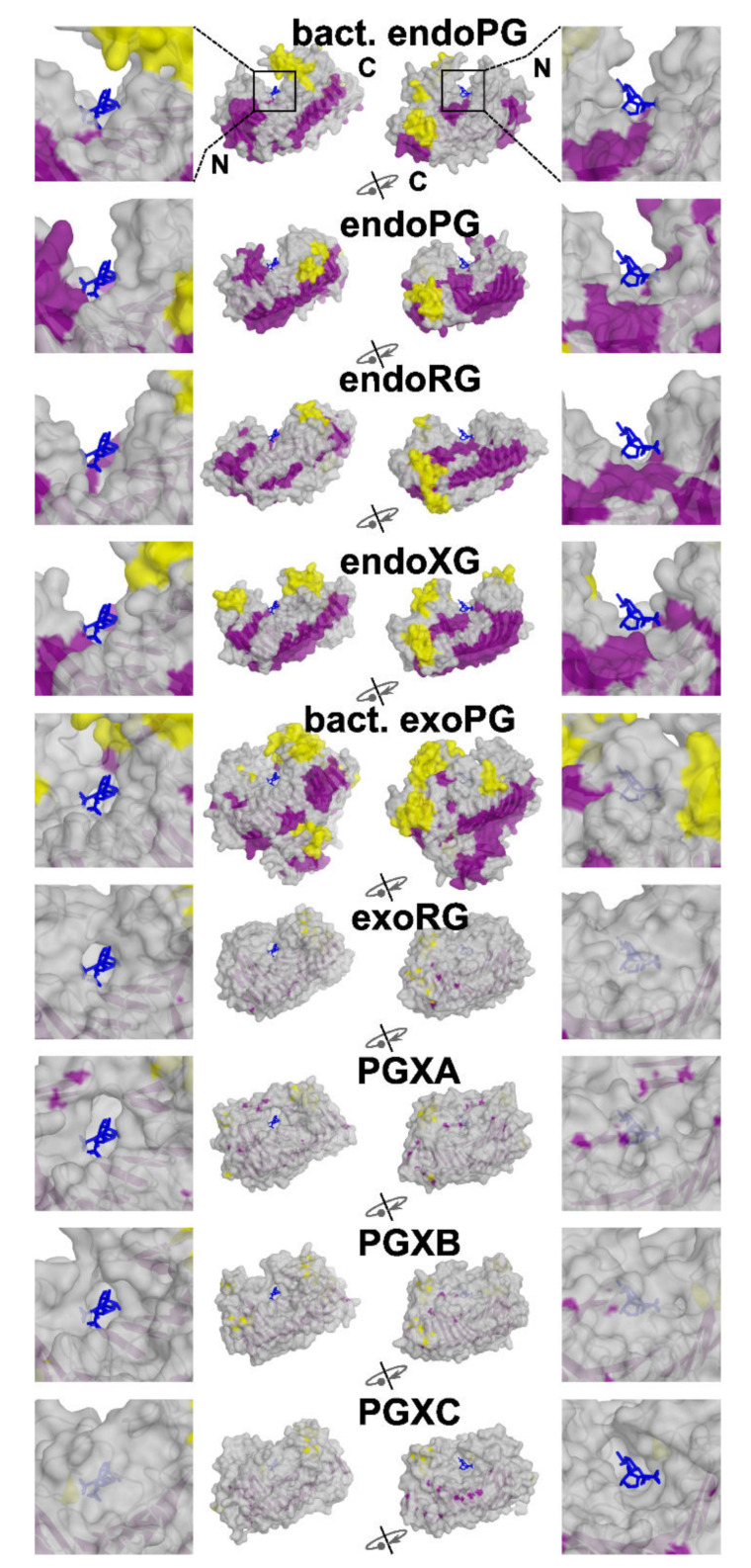
Exomode GH28 enzymes have additional, surface exposed loops. Structures and structure models were aligned to 1KCD, endoPG co-crystallized with two galacturonate molecules (blue sticks). For each reference protein, two rotational views are shown in surface in the center panels, coloured according to secondary structures (β sheets purple, α helixes yellow, coil, grey). N- and C-terminal ends are indicated for bacterial endoPG. Side panels shows in detail the galacturonate ligands. RMSDs to 1KCD: endoXG (4C2L) = 1.144, endoRG (1RMG) = 1.920, endoPG (1CZF) = 0.945, PGXA (PGLRX_ASPFU) = 0.896, PGXB (Q7SAI8_NEUCR) = 1.472, PGXC (A0A1L9N956_ASPTC) = 0.998, exoRG (A0A194WYS5_9HELO) = 0.908, bacterial exoPG (2UVE) = 1.641, bacterial endoPG (1BHE) = 1.791.

**Figure 5 jof-08-00217-f005:**
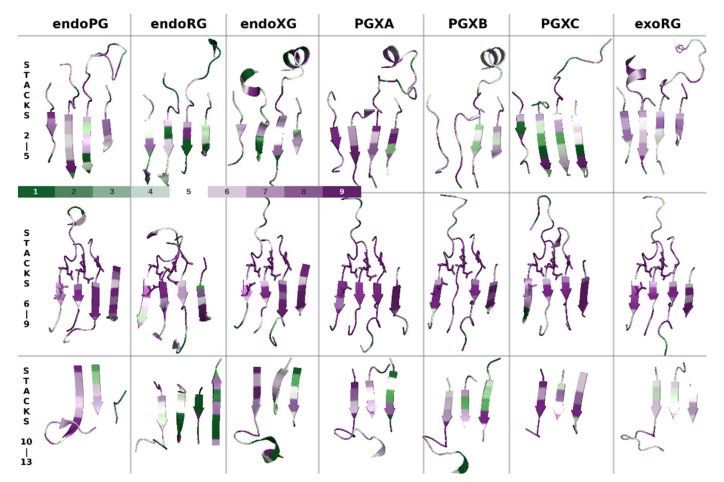
Consurf conservation analysis shows different conservation patterns for different GH28 subfamilies. Shown are selected regions that cover PB1, with some adjacent loop regions, of, in columns, structures 1CZF (endoPG), 1RMG (endoRG) and 4C2L (endoXG), as well as Alphafold structure models for *PGLRX_ASPFU* (*PGXA*), *Q7SAI8_NEUCR* (*PGXB*), *A0A1L9N956_ASPTC* (*PGXC*) and *A0A194WYS5_9HELO* (*exoRG*) aligned to 1CZF. In rows are shown the N-terminal, central and C-terminal regions, conceptually corresponding to subsites −5 to −2, the site with subsite −1 and +1 and subsite +2 and +3, for each of the seven subfamilies. The color scale indicates levels of conservation, 1 being low and 9 being high.

**Figure 6 jof-08-00217-f006:**
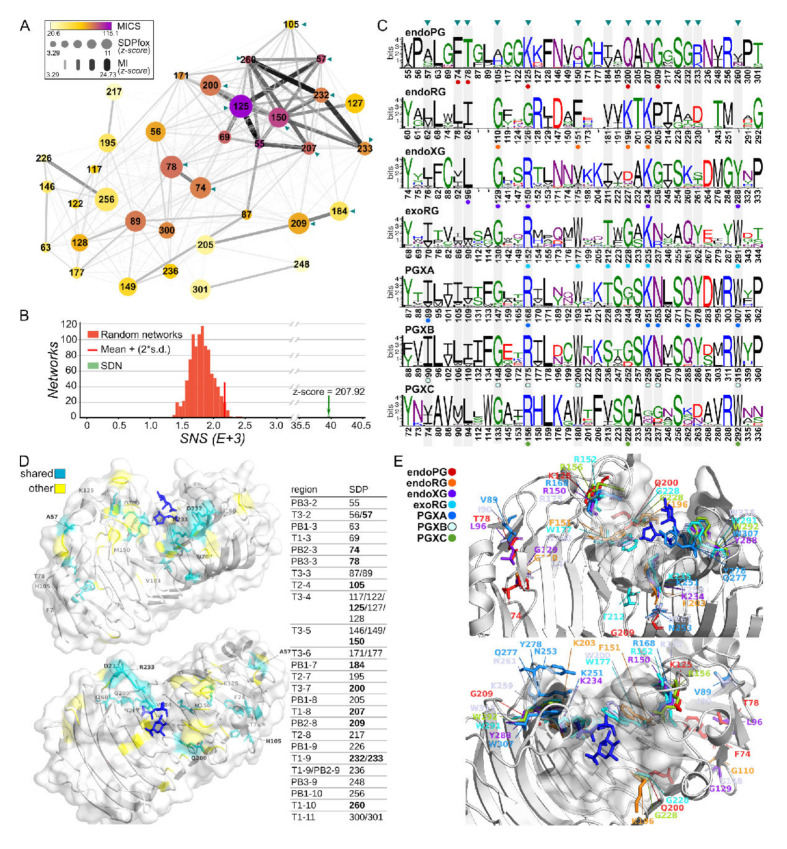
SDP identification and mapping in ascomycete GH28. (**A**) The SDN of the seven-cluster comparison (endoPG vs. endoRG vs. endoXG vs. exoRG vs. PGXA vs. PGXB vs. PGXC) has 36 SDPs represented as nodes with diameter according to SDPfox z-score and color according to its MICS. Edges represent MI z-scores as indicated, numbering according to 1CZF. Triangles indicate shared SDPs, i.e., identified in all four principal analyses (see main text); (**B**) Z-score distribution of 1000 randomly generated networks (incidence histogram) and the SDN (indicated by green arrow). The red vertical line indicates mean + (2*s.d.); (**C**) SDP logos for each class. Numbering according to the subfamily’s reference sequence (described in the main text). Triangles indicate shared SDPs; (**D**) 1CZF in two rotational views with the shared SDPs in light blue sticks, other SDPs are in yellow. Blue sticks, galacturonates (1KCD). The location of the SDPs in the corresponding pleated sheets or turns is indicated in the table, numbering according to 1CZF (bold, shared SDPs); (**E**) 1CZF (cartoon) and other reference structures aligned to 1KCD in two rotational views (galacturonates in blue sticks) with SDPs (sticks) that are highly conserved within their subfamily. Residues are colored and numbered according to the subfamily and its reference sequence, as indicated. Highlighted residues are indicated in (**C**) with correspondingly colored circles.

**Figure 7 jof-08-00217-f007:**
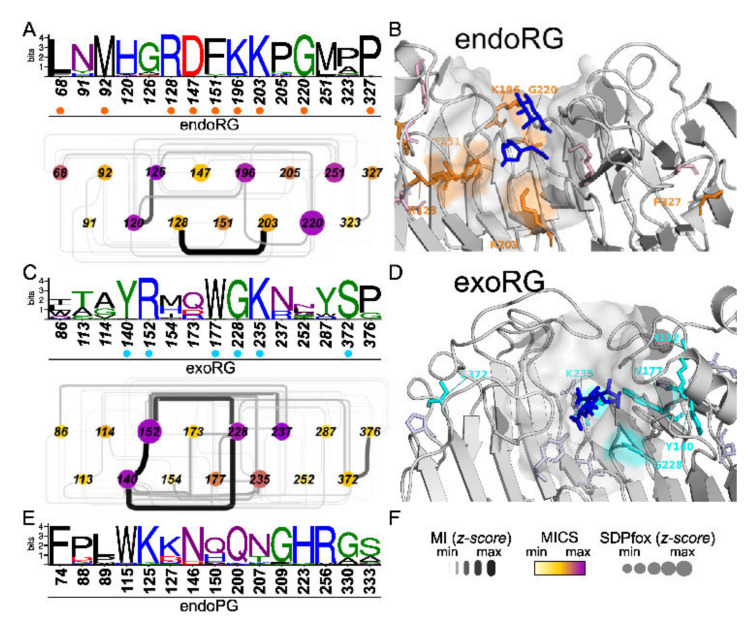
SDP identification in ascomycete endoRG and exoRG classes. (**A**,**C**) Sequence logos and subSDN of SDPs identified in the analyses endoPG vs. endoRG and endoPG vs. exoRG. Numbers correspond to the reference sequence of each subfamily: endoRG and exoRG. SDN-node colors represent the node’s MICS and the diameter corresponds to the SDPfox z-score. Edges thickness and darkness represent MI z-scores (see panel (**F**) for scales). (**B**,**D**) The identified SDPs were mapped on endoRG (**B**) and exoRG (**D**) reference structures aligned to 1KCD (galacturonate substrates in blue sticks). Highly conserved residues with physicochemical differences are labelled and indicated in orange (endoRG) or cyan (exoRG). The surface region corresponding to the binding groove is also shown. (**E**) Sequence logo for the identified SDPs in the endoPG set (numbers correspond to 1CZF).

**Figure 8 jof-08-00217-f008:**
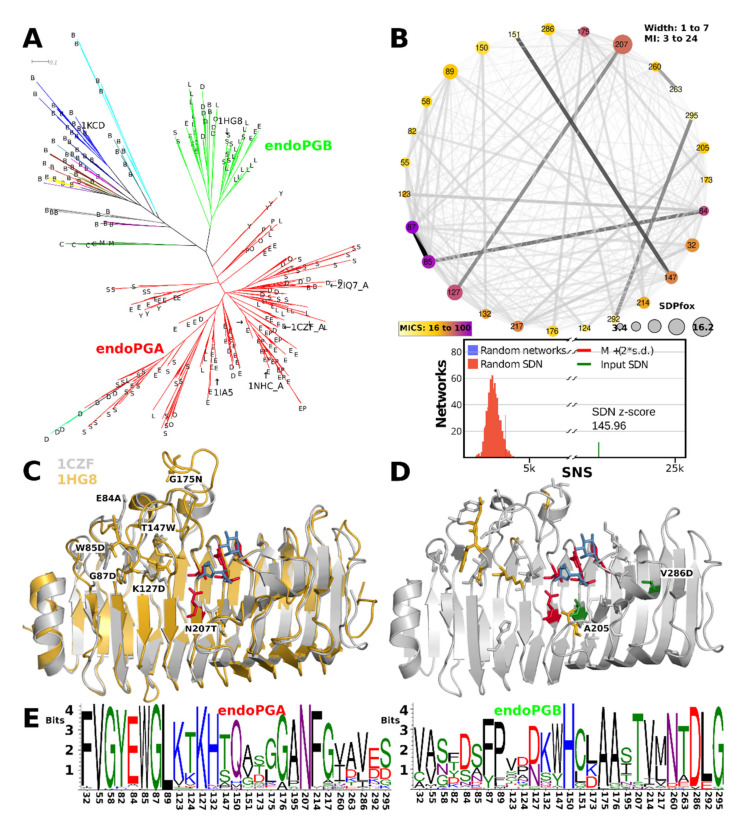
Functional diversification of the ascomycete endoPGA and endoPGB subfamilies involves T3 loops that cover N-terminal subsites. (**A**) Fungal endoPG tree clustered with HMMERCTTER. The taxonomic distribution is indicated by letters: B: basidiomycete; C: chytridiomycete; D: dothideomycete, E: eurotiomycete; L: leotiomycete; M: mucorormycete; O: orbiliomycete; P: pezizomycete; S: sordariomycete; Y: saccharomycete. EP indicates sequences that have Arg91 as processive endoPG 1NHC. Structures are indicated by their PDB identifier. Clade 1 (red) and clade 2 (light green) are mostly ascomycete sequences. A small dark green clade has mucoromycete and chytridiomycete sequences, whereas the remaining 66 clusters are mostly basidiomycete sequences. The sole arrow indicates the common ancestor that determines the subclade with Arg91 sequences but excludes closely related 1CZF; (**B**) SDN of endoPGA vs. endoPGB comparison. The graph shows the SNS compared with that of 1000 random SDNs; (**C**,**D**) SDP mapping. (**C**) 1CZF and 1HG8 are aligned to 1KCD to show the approximate location of the substrate (galacturonates in blue stick only). Catalytic residues are in red stick. Six of the seven SDPs (stick) with the highest MICS locate on T3-3 (84, 85, 87), T3-4 (127), T3-5 (147) and T3-6 (175). SDP204 is on PB1-8, near the catalytic site; (**D**) 1CZF in grey cartoon with top SDPs from C in gold stick, two PB1 SDPs in green stick and the remaining SDPs in grey stick; (**E**) SDP logos with 1CZF numbering.

**Figure 9 jof-08-00217-f009:**
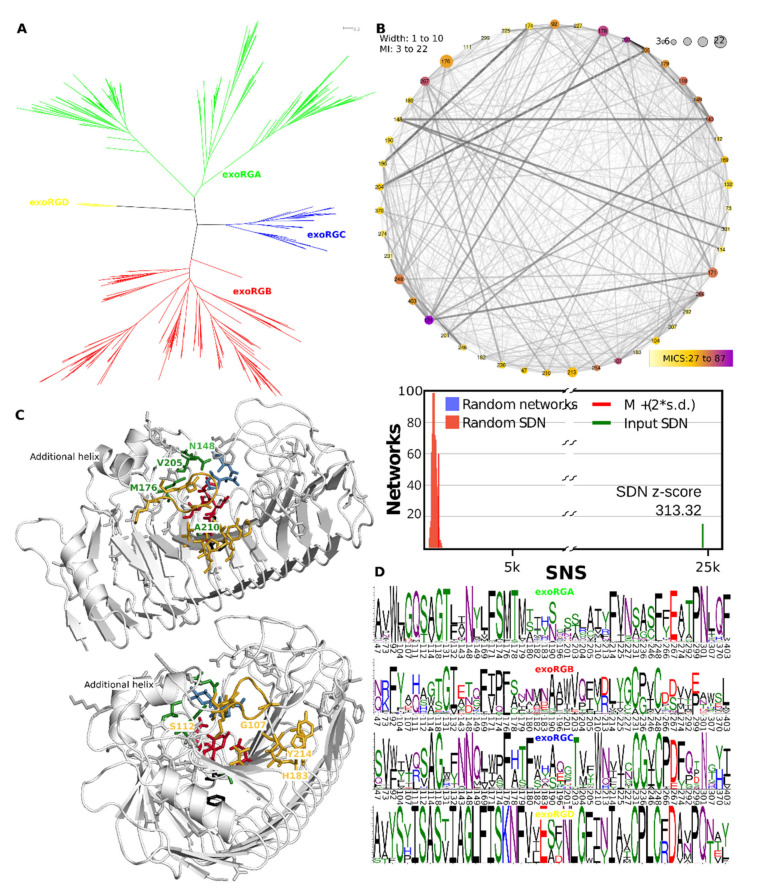
Functional diversification of the ascomycete exoRG subfamilies involves T3 loops that cover subsite −1. (**A**) Fungal exoPG tree clustered with HMMERCTTER; (**B**) SDN of exoRGA/exoRGB/exoRGC comparison. The graph shows the Specific Network Score (SNS) compared with 1000 random SDNs; (**C**) SDP mapping in two different angles. The structure model of exoRG A0A194WYS5_9HELO (grey cartoon) is aligned to 1KCD to show the approximate location of the substrate (galacturonates in blue stick). Catalytic residues are in red stick. Highlighted in green are the residues that appear close to the site. Highlighted in gold are the residues that appear close to subsite −1 where supposedly the rhamnose-moiety binds. In black stick the counterpart of disulphide bridge SS2 that is lacking in exoRGA; (**D**) SDP logos with A0A194WYS5_9HELO numbering.

**Table 1 jof-08-00217-t001:** Classification of fungal GH28 homologues. Indicated are numbers (n) of subfamily homologues per taxon and the proportion with respect to the total number (N) of homologues. NA: Not applied (sets with N < 100). Classification is based on the tree topology, not HMMERCTTER classification.

	All	endoPG		endoRG		endoXG		PGXA		PGXB		PGXC		Mucoro		exoRG		N
Dothideomycete	298	109	0.37	36	0.12	7	0.02	51	0.17	14	0.05	8	0.03	0	NA	73	0.24	298
Eurotiomycete	1150	394	0.34	246	0.21	81	0.07	100	0.09	81	0.07	54	0.05	0	NA	194	0.17	1150
Leotiomycete	429	140	0.33	60	0.14	54	0.13	59	0.14	25	0.06	19	0.04	0	NA	72	0.17	429
Neolectomycetes	2	2	NA	0	NA	0	NA	0	NA	0	NA	0	NA	0	NA	0	NA	2
Orbiliomycetes	30	12	NA	0	NA	4	NA	9	NA	3	NA	0	NA	0	NA	2	NA	30
Pezizomycetes	15	6	NA	4	NA	0	NA	2	NA	3	NA	0	NA	0	NA	0	NA	15
Saccharomycetes	13	13	NA	0	NA	0	NA	0	NA	0	NA	0	NA	0	NA	0	NA	13
Sordariomycetes	787	241	0.31	78	0.10	35	0.04	117	0.15	137	0.17	44	0.06	0	NA	135	0.17	787
Fungi 14919	12	3	NA	1	NA	2	NA	2	NA	1	NA	0	NA	0	NA	3	NA	12
∑Ascomycetes	2736	920	0.34	425	0.16	183	0.07	340	0.12	264	0.10	125	0.05	0	NA	479	0.18	2736
Basidio	499	208	0.42	59	0.12	8	0.02	3	0.01	0	0	100	0.20	0	NA	121	0.24	499
Mucoro	75	2	NA	1	NA	0	NA	0	NA	0	NA	0	NA	72	NA	0	NA	75
Chytridio	9	6	NA	1	NA	0	NA	0	NA	2	NA	0	NA	0	NA	0	NA	9
∑	3319	1136	0.34	486	0.15	191	0.06	343	0.1	266	0.08	225	0.07	72	NA	600	0.18	3319

## Data Availability

All relevant data are in Appendix A.

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
