# Peer review of "Functional Classification and Characterization of the Fungal Glycoside Hydrolase 28 Protein Family"

_jof, 2022, doi:10.3390/jof8030217_

Round 1

Reviewer 1 Report

Please find my comments below:

  • Affiliation needs to be completed. There are no data from research centers.
  • The sentence “The GH28 gene family encodes a number of different enzymatic activities directed at the breakdown of the major plant cell wall component pectin” is unnecessary. Please delete it.
  • First paragraph on page 1- please complete the references.
  • Do the words “Pectate Lyase”, "Rhamnogalacturonan exolyase" and "Rhamnogalacturonan endo-lyase" have to be capitalized?
  • Page 3: Please add the full name of B. cinerea and A. niger.
  • Page 3: The sentence: “It has four parallel β-sheets PB1, PB1a, PB2 and PB3 that together form a β-helix. Each full helix can be seen as a stack or slice of which 1RMG has 14.” Could you unify the size of font?
  • Page 5: Why you use the name “Benen and coworkers [30]”. The first author of this manuscript is Pages.
  • Page 6: What does it mean” (see for instance [45] and other publications in the same special issue of Proteins)”. Could you add one or more references?
  • Page 13: “It is also noteworthy that both α-helices and β-sheets are more exposed to the surface in endo-subfamilies.” Could you unify the size of font?
  • Page 24: What does “Iso213” mean? Perhaps it should be isoleucine (Ile)?

Reviewer 2 Report

In recent years, there has been an increase in interest in sourcing environmentally friendly energy obtained from biofuels obtained from biodegradable materials. Taking into account the economic aspects, such substrates should be first and foremost cheap and environmentally friendly and the obtained biofuels such as biogas or bioethanol, prevent growth
environmental problems.

The systematics of cellulolytic enzymes is quite problematic. Initially, the main criterion classification of cellulolytic enzymes, adopted by Enzymatic Commission (EC) and Nomenclature Committee of the International Union of Biochemistry and Molecular Biology (IUB), was the type of substrate and the way the enzyme interacts with the substrate. These enzymes are grouped according to the reactions they catalyzed, e.g. cellulases, xylanases, chitinases. That was the classification insufficient. Another way to organize the cellulases is by sorting them out on the basis of similarity amino acid sequence. There is usually a dependency between amino acid sequence, fold enzyme and protein tertiary structure. This classification is being updated, however, due to its large size there are problems with the number of hydrolases studied and discovered with current update. In the current update from 2001, glycoside hydrolases are grouped in 86 families. Such a grading system provides valuable information and supplements the data collected by IUB. In 1998 developed and proposed another taxonomy for hydrolases, in which the first three letters indicate the desired and the most favorable substrate, the following numbers are intended to designate the family of hydrolases, and the following letters represent the order in which the cellulases were first described. An example may be be the enzymes found in Trichoderma reesei: CBHI, CBHII and EGI, which are respectively designated Cel7A (CBHI), Cel6A (CBHII) and Cel6B (EGI). In case of, when there is more than one catalytic domain in enzyme, this also translates to names such as Cel9A - Cel48A for the two CelA cellulase catalytic domains from Caldocellulosiruptor saccharolyticus. System this one, however, has some drawbacks, such as instability and hesitations as to the proposed nomenclature, and lack of substrate specificity for exo- and endoglucanases, therefore this system is not yet fully accepted.

There is no description of the cellulosomes
Lack of description of biochemical, immunochemical, ultrastructural and genetic techniques, it became possible to learn about the structure of these proteins.
The authors do not explain the arrangement of the catalytic modules in the scaffoldin, their composition and synergistic effects are still poorly understood.
Regulation of gene expression and production of cellulases are not described
depends on the basis of the microorganisms.
All this needs to be corrected without this the article is not suitable for publication !!!

This article is weak

What are the affiliations of the authors?

Reviewer 3 Report

In the present manuscript, ‘Functional Classification and Characterization of the Fungal Glycoside Hydrolase 28 Protein Family’, the authors have performed a systematic computational analysis of the GH28 superfamily in fungi and also propose a comprehensive nomenclature for particularly ascomycete GH28 enzymes. The concept behind the present study is sound. However, a major revision is suggested for the current manuscript.

  1. Even the authors have not provided the line numbers in the manuscript therefore it is complicated to highlight the section for correction.
  2. The ‘Introduction’ section must include the challenges and bottlenecks in the study of Fungal Glycoside Hydrolase 28 family proteins.
  3. The benefits of PGs present in the human microbiome should be elaborated.
  4. The important and novel findings of this study should be included in the ‘Conclusion’ section.

Round 2

Reviewer 2 Report

The authors did not take into account any of my comments. I do not accept this mmanusktypu for publication.